# Assessing the dynamic impacts of non-pharmaceutical and pharmaceutical intervention measures on the containment results against COVID-19 in Ethiopia

Hongli Zhu[1], Shiyong Liu [2]*, Wenwen Zheng[3], Haimanote Belay[1,4], Weiwei Zhang[1], Ying Qian [5], Yirong Wu[4], Tadesse Guadu Delele[2], Peng Jia[6,7,8]

**1** Research Institute of Economics and Management, Southwestern University of Finance and Economics, Chengdu, China, **2** Institute of Advanced Studies in Humanities and Social Sciences, Beijing Normal University at Zhuhai, Zhuhai, China, **3** Personal Finance Department, HQ of China Construction Bank, Beijing, China, **4** College of Business and Economics, University of Gondar, Gondar, Ethiopia, **5** Business School, University of Shanghai for Science & Technology, Shanghai, China, **6** Department of Public Health, College of Medicine & Health Science, University of Gondar, Gondar, Ethiopia, **7** School of Resources and Environmental Science, Wuhan University, Wuhan, China, **8** International Institute of Spatial Lifecourse Epidemiology (ISLE), Wuhan, China

* shiyongliu2006@gmail.com, liushiyong@bnu.edu.cn

## Abstract

The rapid spread of COVID-19 in Ethiopia was attributed to joint effects of multiple factors such as low adherence to face mask-wearing, failure to comply with social distancing measures, many people attending religious worship activities and holiday events, extensive protests, country election rallies during the pandemic, and the war between the federal government and Tigray Region. This study built a system dynamics model to capture COVID-19 characteristics, major social events, stringencies of containment measures, and vaccination dynamics. This system dynamics model served as a framework for understanding the issues and gaps in the containment measures against COVID-19 in the past period (16 scenarios) and the spread dynamics of the infectious disease over the next year under a combination of different interventions (264 scenarios). In the counterfactual analysis, we found that keeping high mask-wearing adherence since the outbreak of COVID-19 in Ethiopia could have significantly reduced the infection under the condition of low vaccination level or unavailability of the vaccine supply. Reducing or canceling major social events could achieve a better outcome than imposing constraints on people's routine life activities. The trend analysis found that increasing mask-wearing adherence and enforcing more stringent social distancing were two major measures that can significantly reduce possible infections. Higher mask-wearing adherence had more significant impacts than enforcing social distancing measures in our settings. As the vaccination rate increases, reduced efficacy could cause more infections than shortened immunological periods. Offsetting effects of multiple interventions (strengthening one or more interventions while loosening others) could be applied when the levels or stringencies of one or more interventions need to be adjusted for catering to particular needs (e.g., less stringent social distancing measures to reboot the economy or cushion insufficient resources in some areas).

**Data Availability Statement:** All relevant data are within the paper and its Supporting Information files.

**Funding:** The study was supported by the MOE (Ministry of Education in China) Project of Humanities and Social Sciences (21YJAZH053). The funders had no role in study design, data collection and analysis, decision to publish, or preparation of the manuscript.

**Competing interests:** The authors have declared that no competing interests exist.

# 1 Introduction and background

Since the emergence of the first case of COVID-19 on 13 March 2020 in Ethiopia, as of 6 January 2022, this country had reported 436,586 confirmed cases and 6,988 deaths [1]. Furthermore, during the same period, due to the global COVID-19 vaccine inequity [2, 3], there were only 1.4% and 6.6% of the total population being fully vaccinated and partially vaccinated, respectively [4, 5].

As one of the ancient countries with strong religious people, in Ethiopia, 43.5% of the population are Orthodox Christians, 33.9% of the population are Muslims, and the rest are protestants and traditional religious followers [6]. Ethiopian Orthodox Churches encourage the communities to pray more in groups together by reinforcing a feeling of unity in its people. In Orthodox Church, people made greetings to each other by shaking hands and having cheek-to-cheek kisses, and face masks were not used in the church because the church is deemed a sacred place [6]. Muslim communities also have similar greeting etiquettes. Such experiences were critical contributing factors to the increased risk of COVID-19 transmission. Moreover, Ethiopia is the second-most populous country in Africa next to Nigeria, and it is prevalent for people to live together with their extended families under one roof, eat together from one plate daily, and move in groups on the very narrow paved road since this country has solid social solidarity [7]. Therefore, the practices mentioned above of religion, cultural, and social interactions in Ethiopia posed significant challenges in effectively containing and controlling COVID-19.

In the meantime, Ethiopia is one of the poorest countries in the world, with a per capita GDP of $936.34 in 2020 [8]. As of 2021, nearly 23% of the population lives in extreme poverty [9]. A study confirmed that hospital preparedness in the selected state was tremendously insufficient as per World Health Organization measurement, with one out of eight hospitals admitting COVID-19 patients [10]. What is more serious is that, according to statistics from the World Bank, the standard hospital beds per thousand people ratio of 0.33 in 2016 in Ethiopia was far from the WHO minimum standard (3 beds per 1000) [11], which rendered the unadmitted COVID-19 patients helpless and forced them to become the source of infection.

With the advent of COVID-19, scientists in public health worldwide have been trying to investigate the gaps and challenges in the containment measures and, consequently, evaluate a particular country's preparedness for the new emerging disease [12–15]. As non-pharmaceutical interventions (NPIs) such as social distancing, hand washing, and face mask-wearing have been proved effective in providing necessary help, scholars have examined the knowledge, attitude, and practices of face mask utilization affecting them in Ethiopia [16–21]. Ayele Tadesse Awoke et al.'s survey in the Amhara region of Ethiopia revealed that, at the early stage of the COVID-19 pandemic, levels of adherence regarding hand hygiene, physical distancing, and mask utilization were 12.0%, 13.00%, and 26%, respectively, which demonstrated regional variation. They suggested that community-based education would increase the practices mentioned above [22]. The surveys conducted by Endriyas et al. and Haftom & Petrucka showed that the levels of mask utilization were more than 50% [23, 24]. Ayele Wondimu et al. evaluated the spread dynamics of COVID-19 in Ethiopia under the assumptions of face mask utilization of 20%, 40%, and 60%. They also assessed the combined effects of enforcing social distancing and increasing the adherence to mask-wearing and found significant reductions in infections [25]. Studies from Tucho and Kumsa, and Zewude et al. indicated the challenges of keeping certain compliance levels of mask-wearing [26, 27]. Ejigu et al.'s research predicted the infections under the assumptions of implementing different NPIs including social distancing, mask-wearing, and sanitary measures [28].

Bushira used geospatial techniques and the CHIME model to evaluate the impacts of 25%, 75%, and 95% social distancing interventions on flattening and delaying the curve [29]. Deressa examined the practices of social distancing of government employees in Addis Ababa and the results exhibited a 96% adherence level to mask-wearing [30]. Fikrie et al.'s study identified that knowledge and attitude were contributing factors causing poor practices of social distancing in the West Guji Zone of Ethiopia [31]. Hailu et al. investigated the barriers and driving factors for influencing the compliance level of social distancing measures [17]. In Tolu et al.'s study, the calculated ReadyScore was 52% indicating that more measures needed to be implemented, where they recommended social distancing measures, increasing case tracing, sanitary measures, etc [15].

Apart from assessing the impacts of NPIs on the spread dynamics in Ethiopia, Suthar et al. evaluated the vaccination (i.e., pharmaceutical interventions-PIs) on the transmission dynamics of COVID-19 [32]. Deressa and Duress used the mathematical epidemiological model to evaluate and identify the optimal combination of multiple measures-i.e., public health education, personal protective measures, and hospitalization of the infected [33].

Researchers have used different models to predict the spread trend of COVID-19 over time. Abebe used an exponential smoothing model for the new COVID-19 infections [34]. Eticha employed a case-based autoregressive integrated moving average model to predict the new cases of COVID-19 [35]. In a study done by Gebretensae and Asmelash, the researchers used Box–Jenkins modeling framework, i.e., ARIMA (p, d, q), to forecast the trend of COVID-19 spread in Ethiopia [36]. Gebremeskel et al. applied a compartmental epidemic model to predict the transmission dynamics of COVID-19 in Ethiopia [37]. Habenom et al. adopted a model with fractional differential equations to analyze the transmission dynamics of COVID-19 in Ethiopia [38].

Given the capability of system dynamics (SD) models to capture nonlinearity between cause and effect and integrate time delays and feedback loops prevalent in disease progression and population health, SD models have been widely used in multiple areas of health-related research since the 1970s [39]. Its broad applications have been seen in studying the dynamics of both infectious diseases [40, 41] and non-communicable chronic diseases [42–52]. SD's applications in health-related fields were also often witnessed in evaluating the impacts of policies and interventions [53–61]. SD models were also widely used in studies related to health service improvement (e.g., hospital management) [62–64]. SD models were well recognized and practiced in research in developing and evaluating national health policy [54, 65–69] and investigating complexity and uncertainties in healthcare and health-related socioeconomic systems [55, 62, 70–75]. In the past two years, scholars have extensively used SD models to understand the transmission dynamics, impacts of containment measures, and prediction of COVID-19 spread [76–87].

Majorities of previous research have focused on the evaluations of limited interventions and their combinations without simultaneously capturing the impacts of NPIs and PIs. For example, a typical SEIR model was used by Ayele Wondimu et al. to assess the impacts of social distancing measures and different compliance levels of face mask-wearing [25]. The SEIR models employed by Ejigu et al. and Taye et al. evaluated the impacts of interventions such as social-distancing measures, mask-wearing, and handwashing [28, 88]. Our extended SEIR further considered the impacts of a lot more factors including hospitalized patients, un-hospitalized patients, patients with mild and severe symptoms, asymptomatic patients, vaccine administration levels, hospital/quarantine hospital capacity, and different levels of compliance in face mask-wearing.

To gain a better understanding of issues that existed in the containment measures against COVID-19 disease at the previous stage and gain insights into the pandemic trend under

different intervention scenarios, this study mainly focuses on the application of the SD model to conduct counterfactual analysis (16 scenarios) to identify issues and gaps in the past containment. It also aims to control measures and capture the possible future transmission dynamics of COVID-19 under various combinations of interventions, including NPIs and PIs (264 scenarios). In the NPI, we evaluated the stringency of social distancing measures (three levels of stringency for social events and routine activities represented by average contact rates-see Table 2 and Appendix C in S1 Appendix [25]), the adherence level of face mask-wearing (three levels with 48% [18, 19] in baseline, 60%, and 70% in the proposed assumptions), and hospital beds (20000 beds in baseline [89, 90]-including temporary beds in isolation centers, 12000–60%, 16000–80%, 40000–200% in the proposed assumptions). Regarding PIs, we considered the capacity of vaccine supply and administration rate (30%- and 20% of the total population being administered the 1st & 2nd dose, 20% and 10% of the total population being administered the 1st & 2nd dose in the proposed assumptions), vaccine efficacy (68.4% for the first dose and 80% for the second dose [91, 92], 40% for the first dose and 60% for the second dose in the proposed assumptions), and immunological period (240 months in the baseline [93–96], proposed assumption of 180 months).

This paper is structured as follows. Section 1 presents the background and literature review for this study. In Section 2, methodology and formations are provided. Section 3 shows the three types of simulation results, including calibration and parameter estimation, counterfactual analysis, and trend analysis under different containment strategies. Section 4 concludes this study with a discussion of the results. A statement on the limitations and opportunities for future research is also discussed.

## 2 Materials and methodology

### 2.1 Context and design for the study

This study intends to investigate the COVID-19 spread dynamics from the emergence of the first case till Nov 5, 2021 (by doing counterfactual analysis) and to predict the transmission trend of the disease till Nov 5, 2022, under different possible containment and control measures in Ethiopia. We built a model in this research attempting to capture factors including COVID-19 characteristics (infectivity, incubation period, fractions of symptomatic and asymptomatic infections, rate of severe case), major social events (religious events, war in Tigray, election campaign), the stringency of containment measures (social distancing measures, adherence level of face mask-wearing), and vaccine (efficacy, supply, and administration capacity, immunological period). The model considers the 114.9 million total population of Ethiopia.

### 2.2 Model structure and formulations

In this study, we built a system dynamics model with the first graph (Fig 1) demonstrating a causal loop diagram that illustrates the feedback loops and causal process tracing of variables and the second graph showing main stock flow variables (Fig 2). Fig B1 in Appendix B (S1 Appendix) also provides a detailed view of the model structure. Built on previous literature, this SD model is an extended SEIR compartmental model which includes state variables as **S(t)**-susceptible population, **E(t)**-exposed population, **SYC(t)**-symptomatic patients, **AYC(t)**-asymptomatic patients, **CM(t)**-confirmed mild cases, **SC(t)**-severe cases, **D(t)**-deaths, and **R(t)**-recovered cases. The model also captures structures related to untreated cases and vaccination, which has **UCM**(t)-untreated mild cases, **USC**(t)-severe untreated cases, **UR**(t)-untreated recovered cases, **UD**(t)-untreated deaths, **SV1**(t)-population administered 1st dose vaccine, **SV2**(t)-population administered 2nd dose vaccine, **TA1**(t)-total available 1st dose vaccine reserve, and **TA2**(t) -total available 2nd dose vaccine reserve.

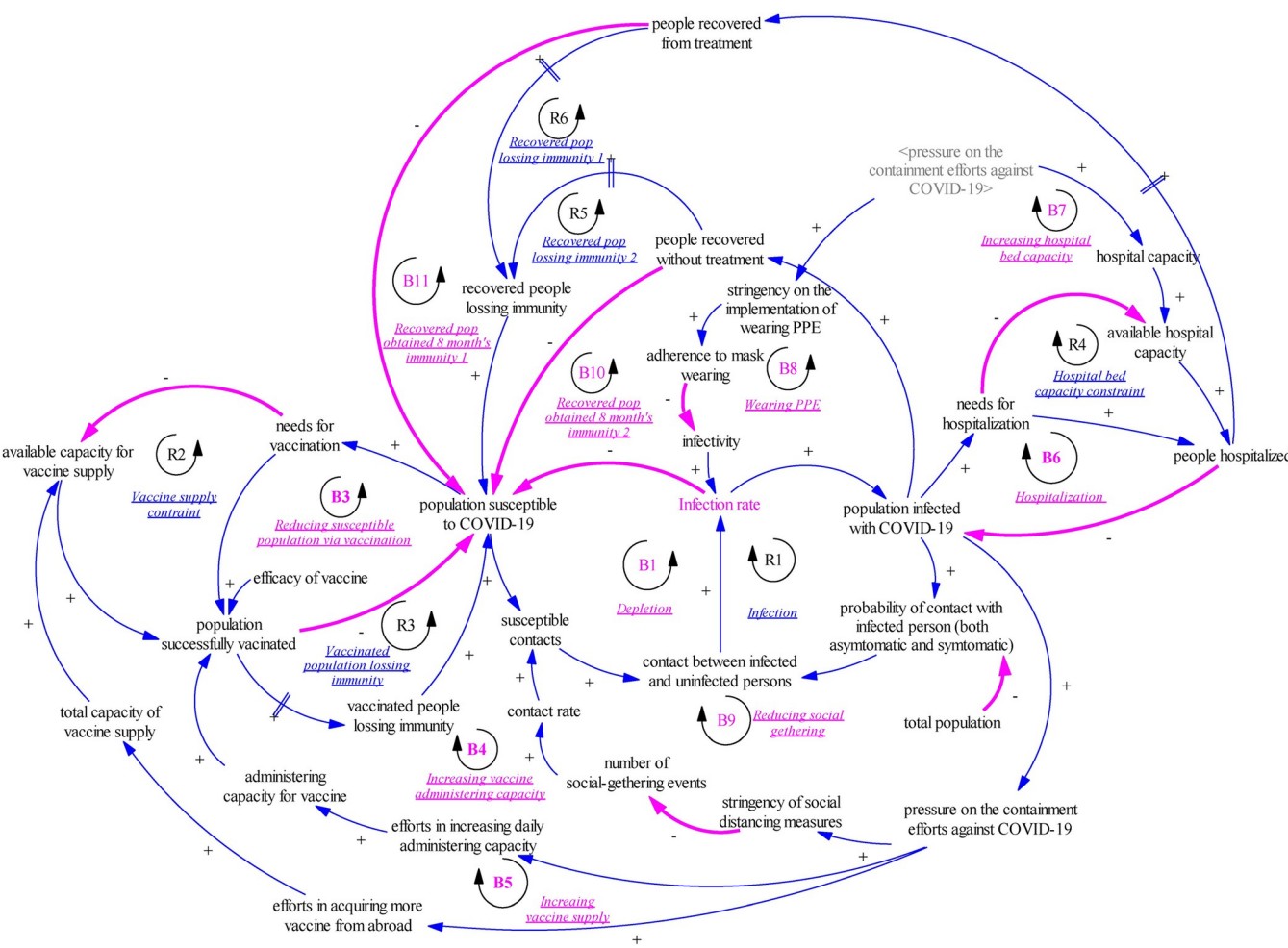

**Fig 1. Causal loop diagram for the system dynamics model.**

## 2.3 Data collection and quality

The COVID-19-related infections, recovery, deaths, and vaccination data were obtained from online published reports of the World Health Organization and Ministry of Health-Ethiopia [1, 4] (https://covid19.who.int/region/afro/country/et; https://www.trade.gov/country-commercial-guides/ethiopia-healthcare; https://covid19.healthdata.org/ethiopia?view=resource-use&tab=trend&resource=all_resources). Data and parameters relate to average contact rate [25], the infectivity of both symptomatic and asymptomatic cases [97, 98], the ratio of symptomatic cases [99, 100], incubation period [101, 102], and time from severe symptomatic to recovery (treatment) [99, 103] were all retrieved from literature and confirmed in calibration. As for the setting and calibrations for other parameters, please refer to Table 1 for details.

# 3 Results

## 3.1 Model calibration

The model was calibrated against official data obtained from WHO and the Ministry of Health-Ethiopia by covering the period from March 13th, 2020, to November 5th, 2021, which

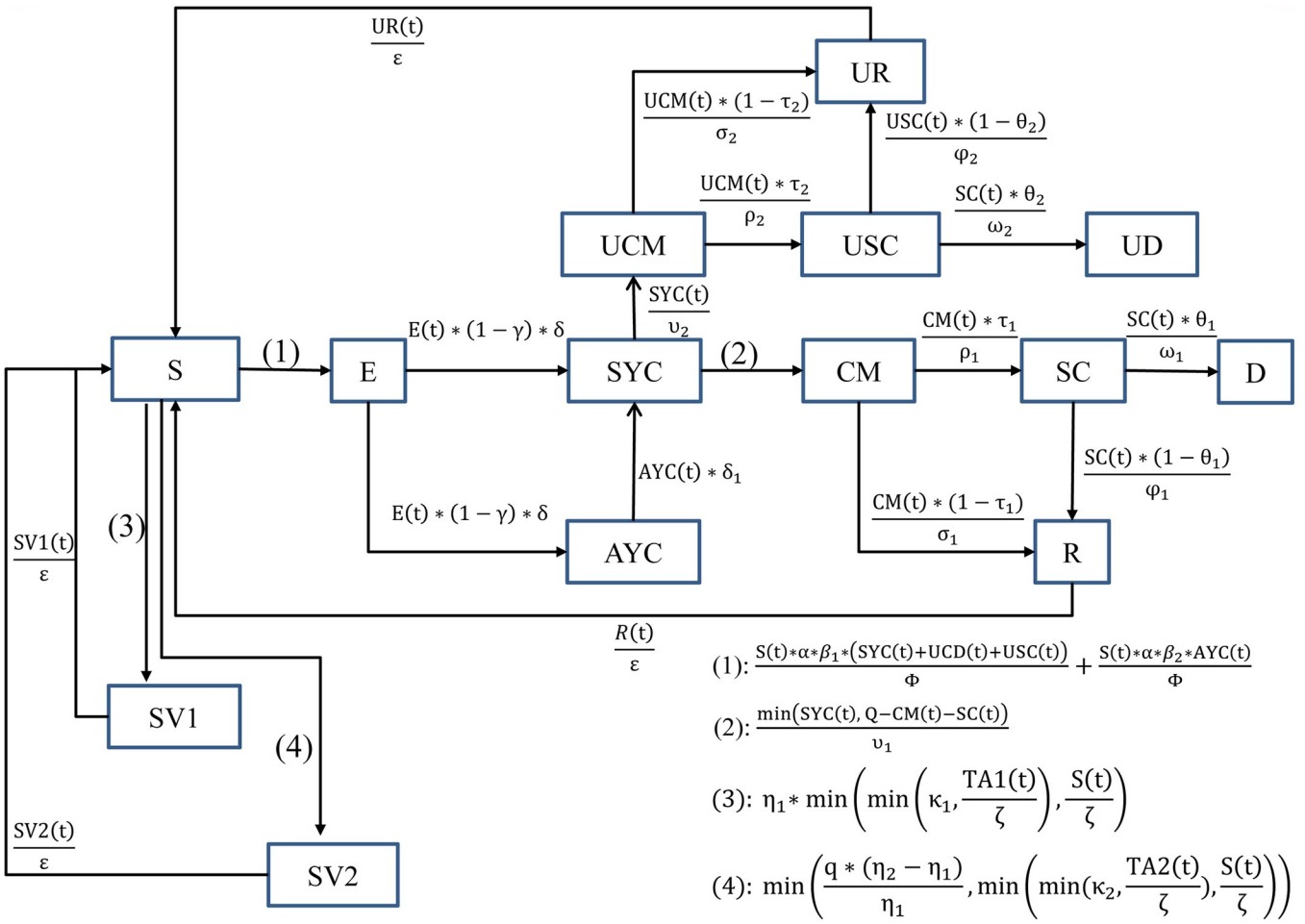

**Fig 2. Simplified stock flow diagram for the Ethiopia SEIR model.**

is a total of 603 days (refer to Fig 3) [1]. The estimated parameters listed in Table 1 were aligned with references and expert inputs.

Based on the calibrated model, simulation scenarios in the following section were designed by assuming different values in relevant parameters-i.e., medical resources (mainly considering beds), level of mask-wearing adherence, vaccine supply, doses of vaccine administered, efficacies of 1st and 2nd dose of vaccine, immunological period, average contact rates under the circumstance of having social events and imposition of social distancing measures (SDMs). The level of mask-wearing adherence was also used to calculate weighted symptomatic and asymptomatic infectivity (refer to Table 2). Counterfactual analysis and future transmission trend analysis were then conducted to gain a better understanding of the impacts of different interventions to inform policy development and decision-making.

## 3.2 Counterfactual analysis

The counterfactual analysis covered the period from March 13th, 2020, to November 5th, 2021. In Fig 4, for all four scenarios 1, 4, 9, and 12, the most significant impacts were from the measure of imposing social distancing on routine activities, which could have reduced the cumulative infections from more than 360,000 to less than 240,000. Under the same circumstance, the

**Table 1. Model calibration results.**

| Variable | Variable name | Initial value | Data sources | Units |
|---|---|---|---|---|
| $S(t)$ | Susceptible population | 1.14964e+008 | Empirical data | Person |
| $E(t)$ | Exposed population | 0 | Empirical data | Person |
| $SYC(t)$ | symptomatic case | 1 | Empirical data | Person |
| $AYC(t)$ | asymptomatic case | 0 | Empirical data | Person |
| $CM(t)$ | confirmed mild case | 0 | Empirical data | Person |
| $SC(t)$ | severe case | 0 | Empirical data | Person |
| $D(t)$ | Death | 0 | Empirical data | Person |
| $R(t)$ | recovered population | 0 | Empirical data | Person |
| $UCM(t)$ | Untreated confirmed mild case | 0 | Empirical data | Person |
| $USC(t)$ | Untreated severe case | 0 | Empirical data | Person |
| $UD(t)$ | Untreated death population | 0 | Empirical data | Person |
| $UR(t)$ | Untreated recovered population | 0 | Empirical data | Person |
| $SV1(t)$ | Successfully Vaccinated population with 1st Dose | 0 | Empirical data | Person |
| $SV2(t)$ | Additional successfully vaccinated population with 2nd dose | 0 | Empirical data | Person |
| $TA1(t)$ | Total Available 1st Dose Vaccine Reserve | 0 | Empirical data | dose |
| $TA2(t)$ | Total Available 2nd Dose Vaccine Reserve | 0 | Empirical data | dose |
| $Q$ | national hospital capacity (CoVID-19) | 20000 | Empirical data | bed |
| $\Phi$ | Total population | 1.14964e+008 | Empirical data | Person |
| $\alpha$ | contact rate per day | Appendix 1 | [25] and calibrated | Person/day |
| $\beta_1$ | Infectivity of symptomatic cases | Appendix 1 | [97, 98] and calibrated | Dmnl |
| $\beta_2$ | infectivity of asymptomatic cases | Appendix 1 | [97, 98] and calibrated | Dmnl |
| $\gamma$ | ratio from Exposed to symptomatic | 0.95 | [99, 100] and calibrated | Dmnl |
| $\delta$ | incubation period | 5.2 | [101, 102] | day |
| $\delta_1$ | time from asymptomatic to symptomatic | 14–5.2 = 8.8 | [99] | day |
| $\upsilon_1$ | diagnosis time | 0.321721 | calibrated | day |
| $\upsilon_2$ | time from symptomatic cases to untreated mild cases | 3 | calibrated | day |
| $\rho_1$ | time from confirmed mild symptomatic to severe symptomatic (treatment) | 3 | calibrated | day |
| $\sigma_1$ | recovery time from mild to recovery (treatment) | 15 | calibrated | day |
| $\tau_1$ | ratio from mild to severe | 0.0117848 | calibrated | Dmnl |
| $\varphi_1$ | time from severe symptomatic to recovery (treatment) | Appendix 1 | [99, 103] and calibrated | day |
| $\omega_1$ | time from severe symptomatic to death (treatment) | Appendix 1 | calibrated | day |
| $\theta_1$ | death ratio (treatment) | Appendix 1 | calibrated | Dmnl |
| $\rho_2$ | time from mild symptomatic to severe case (untreated) | 7.28445 | calibrated | day |
| $\sigma_2$ | the recovery period from untreated mild cases | 17 | calibrated | day |
| $\tau_2$ | ratio from untreated mild symptomatic to severe symptomatic | 0.0919931 | calibrated | Dmnl |
| $\varphi_2$ | The recovery period from untreated severe cases | Appendix 1 | calibrated | day |
| $\omega_2$ | The period from untreated severe Symptomatic to death (untreated) | Appendix 1 | calibrated | day |
| $\theta_2$ | death ratio without treatment | Appendix 1 | calibrated | Dmnl |
| $\lambda_1$ | supply of 1st dose vaccine from various sources | Table 1 Sheet 1 | Empirical data | dose |
| $\lambda_2$ | supply of 2nd dose vaccine from various sources | Table 1 Sheet 1 | Empirical data | dose |
| $\varepsilon$ | average immunological memory period | 240 | Assumption | day |
| $\kappa_1$ | actual administering capacity of 1st dose vaccine per day | Table 1 Sheet 2 | Empirical data | dose |
| $\kappa_2$ | actual administering capacity of 2nd dose vaccine per day | Table 1 Sheet 2 | Empirical data | dose |
| $\eta_1$ | effectiveness of 1st dose | 0.684 | Assumption | Dmnl |
| $\eta_2$ | effectiveness of 2nd dose | 0.8 | Assumption | Dmnl |
| $q$ | Delay time in administering 2nd dose vaccine | 14 | Assumption | day |
| $\zeta$ | Time 1 | 1 | calibrated | day |

**Table 2. Scenario settings in the counterfactual analysis.**

| Scenarios | Medical resources | Vaccine supply and administration | | | | Interventions | | | |
|---|---|---|---|---|---|---|---|---|---|
| | Hospital beds | Vaccine administration | Vaccine supply | ✓Vaccine efficacy | Immuno-logical period (days) | † Social event- Average contact rate | †† Imposing social distancing- Average contact rate | *Symptomatic infectivity- weighted by mask- wearing adherence | **Asymptomatic infectivity- weighted by mask-wearing adherence |
| S 1 | 20000 | No change | No change | 68.4%,80% | 240 | Table A1 (S1 Appendix) | 14 | Table A4 (S1 Appendix) | Table A5 (S1 Appendix) |
| S 2 | 20000 | No change | No change | 68.4%,80% | 240 | 22 | Table A1 | Table A4 (S1 Appendix) | Table A5 (S1 Appendix) |
| S 3 | 20000 | No change | No change | 68.4%,80% | 240 | 22 | 14 | Table A4 (S1 Appendix) | Table A5 (S1 Appendix) |
| S 4 | 20000 | No change | No change | 40%, 60% | 240 | Table A1 (S1 Appendix) | 14 | Table A4 (S1 Appendix) | Table A5 (S1 Appendix) |
| S 5 | 20000 | No change | No change | 40%, 60% | 240 | 22 | Table A1 | Table A4 (S1 Appendix) | Table A5 (S1 Appendix) |
| S 6 | 20000 | No change | No change | 40%, 60% | 240 | 22 | 14 | Table A4 (S1 Appendix) | Table A5 (S1 Appendix) |
| S 7 | 20000 | No change | No change | 68.4%,80% | 240 | Table A1 (S1 Appendix) | 14 | 55% | 55% |
| S 8 | 20000 | No change | No change | 40%, 60% | 240 | Table A1 (S1 Appendix) | 14 | 55% | 55% |
| S 9 | 40000 | No change | No change | 68.4%,80% | 240 | Table A1 (S1 Appendix) | 14 | Table A4 (S1 Appendix) | Table A5 (S1 Appendix) |
| S 10 | 40000 | No change | No change | 68.4%,80% | 240 | 22 | Table A1 | Table A4 (S1 Appendix) | Table A5 (S1 Appendix) |
| S 11 | 40000 | No change | No change | 68.4%,80% | 240 | 22 | 14 | Table A4 (S1 Appendix) | Table A5 (S1 Appendix) |
| S 12 | 40000 | No change | No change | 40%, 60% | 240 | Table A1 (S1 Appendix) | 14 | Table A4 (S1 Appendix) | Table A5 (S1 Appendix) |
| S 13 | 40000 | No change | No change | 40%, 60% | 240 | 22 | Table A1 | Table A4 (S1 Appendix) | Table A5 (S1 Appendix) |
| S 14 | 40000 | No change | No change | 40%, 60% | 240 | 22 | 14 | Table A4 (S1 Appendix) | Table A5 (S1 Appendix) |
| S 15 | 40000 | No change | No change | 68.4%,80% | 240 | Table A1 (S1 Appendix) | 14 | 55% | 55% |
| S 16 | 40000 | No change | No change | 40%, 60% | 240 | Table A1 (S1 Appendix) | 14 | 55% | 55% |

Notes

✓ Based on the literature, the assumed efficacies of the first dose and second dose COVID-19 vaccine are 68.4% and 80%, respectively, in the baseline setting. In the counterfactual analysis, we want to see the impact of low vaccine efficacy on the spread dynamics. Of course, we also combine this factor with other proposed changes in the analysis.

† and †† Here, we consider the situations by incorporating social events and imposing social distancing containment measures. In the counterfactual analysis, we use the average contact rates of 22 and 14 for the simulation periods with social events and the average period with enhanced social distancing containment.

* and ** We referred to the mask-wearing data for Ethiopia in the historical data coming from IHME (Institute for Health Metrics and Evaluation) https://covid19.healthdata.org/ethiopia?view=mask-use&tab=trend.

It was assumed that 48% mask-wearing adherence in the baseline settings and 55% mask-wearing adherence in some scenarios of the counterfactual analysis were used.

reduction in cumulative death (40.7%) was more significant than cumulative infections (34.75%). Doubling hospital capacity while keeping all other factors constant could slightly reduce the number of cumulative infections (1.2%, S1 vs. S9). It was because the average number of infections per day did not exceed the hospital capacity, and not all COVID-19 patients

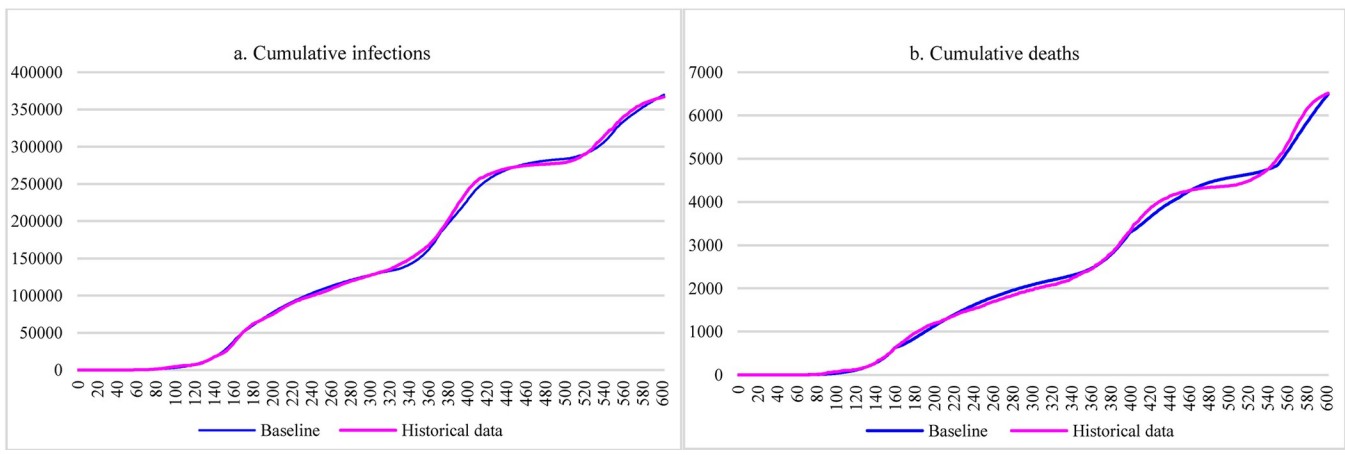

**Fig 3. Baseline results by calibrating the model against historical data.**

could be admitted to the hospital given the accessibility issues of hospitals in some regions, where the impact was a lot smaller than that of the measures of implementing social distancing (34.7% reduction, S1 vs. historical data). Due to the meager vaccination rate (as of November 6, 2021, with 6.07% partially vaccinated and 1.23% fully vaccinated) [4], the reduced efficacy of COVID-19 (S1 vs. S4) could only slightly increase the incidence of infections by 0.088%. Doubling hospital capacity could reduce cumulative deaths by 1.8%, slightly larger than the reduction in accumulative infections.

In Fig 5, lowering the contact rate in different social events (e.g., reducing the scale of those events by canceling the events, or emphasizing social distancing during social events) while holding other factors unchanged could achieve a significant reduction in the cumulative infections (89.01%, S2 vs. Historical data). Due to the significant reduction in cumulative infections, increasing hospital capacity did not reduce cumulative infections and deaths (S2 vs. S10). Similarly, due to the low vaccination rate, the reduced efficacy of the vaccine increased the number of infections by 25 (S2 vs. S5).

In Fig 6, this group of scenarios demonstrates the impacts of enforcing social distancing for main social events and routine life activities. The results showed not much different from

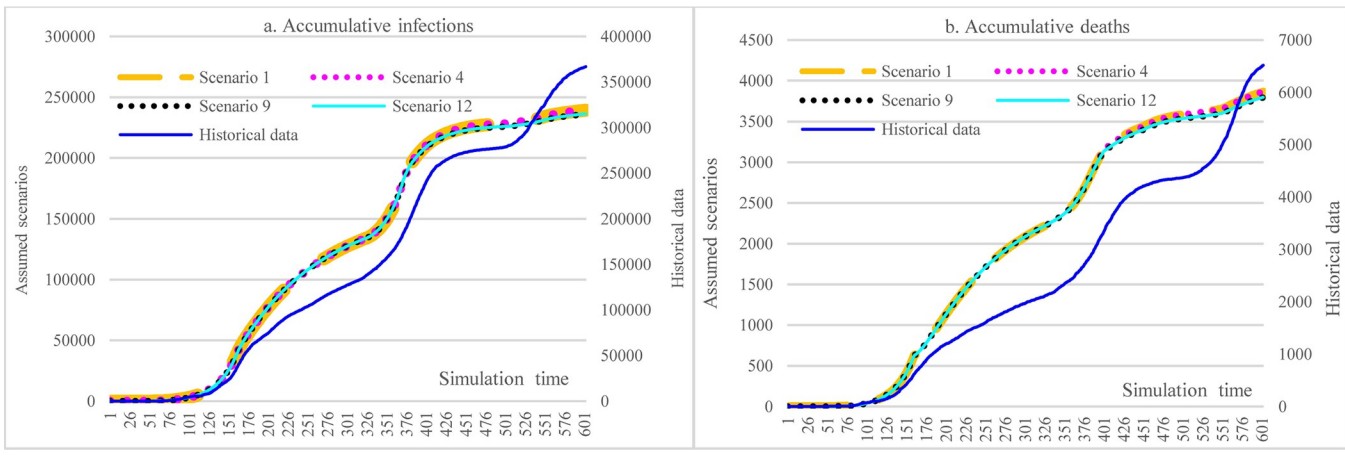

**Fig 4. Impacts of hospital capacity, vaccine efficacy, and social distancing in counterfactual analysis.**

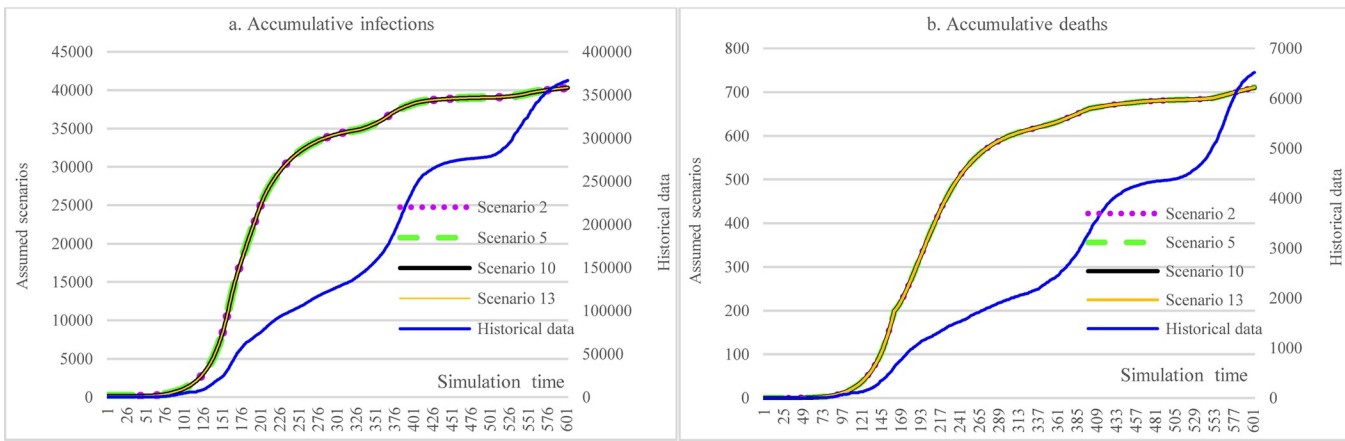

**Fig 5. Impacts of enforced social distancing for social events in counterfactual analysis.**

those of values in Fig 5, which obtained an 89.35% of reduction in cumulative infections (S3 vs. Historical data). Other impacts followed the pattern in Fig 5.

In Fig 7, this group of scenarios assumed that the adherence level to wearing face masks was 55% since the first COVID-19 infections in Ethiopia. The results revealed a considerable reduction in the cumulative infections, less than 1500. Furthermore, the cumulative deaths dropped to 20 in this setting.

In Fig 8, this group of scenarios compared the impacts of enforcing social distancing on either main social events or routine life activities and their combination (S1, S2, S3). Fig 8 shows that enforcing social distancing on major social events can achieve much more impact in reducing infections than enforcing the same containment measures on routine life activities (S1: S2: S3 = 34.75%: 89.01%:89.35%).

## 3.3 COVID-19 transmission dynamics in Ethiopia under different scenarios

In evaluating the impacts of different possible future scenarios on COVID-19 spread in Ethiopia in the coming period, we proposed and simulated 264 scenarios (Refer to Tables C1

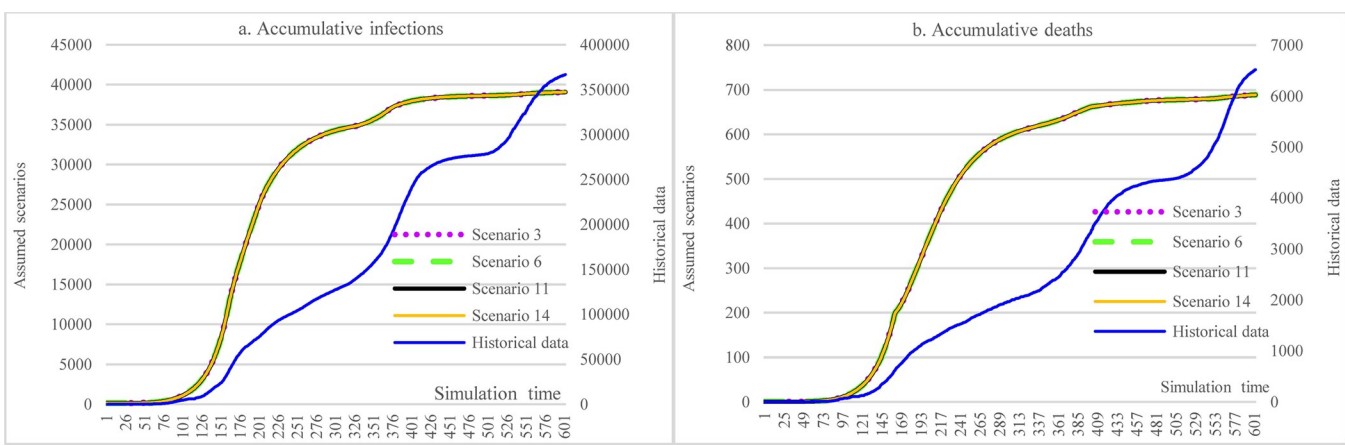

**Fig 6. Impacts of enforced social distancing for both social events and routine life activities in counterfactual analysis.**

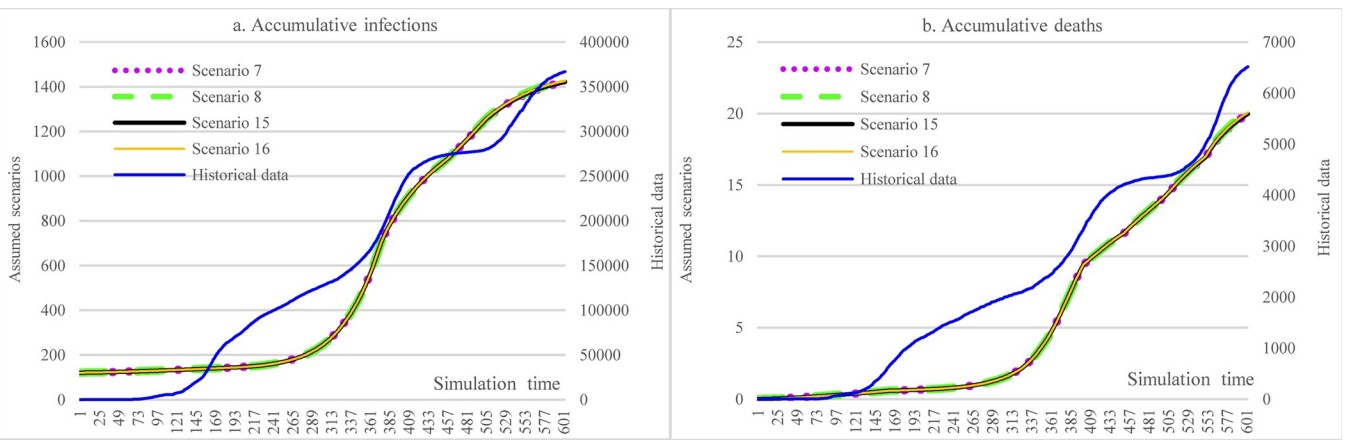

**Fig 7. Impacts of increasing face mask-wearing in counterfactual analysis.**

through C7 in Appendix C (S1 Appendix)). Factors considered in the scenario analysis include hospital capacity (doubling, 80%, and 60% of current capacity), vaccine administration and supply (as is-same as what had been administering; 1st dose 20% population, 2nd dose 10% population; 1st dose 30% population, 2nd dose 20% population), vaccine efficacy (1st dose 68.4%, 2nd dose 80%; 1st dose 40%, 2nd dose 60%), immunological period of vaccine, average contact rates for social events and everyday activities, and adherence level of mask-wearing (48%, 60%, and 70%). The prediction simulation periods covered from November 6th, 2022, to November 5th, 2022. For the sake of increasing visualization effects for comparison among different scenarios, we set the bound minimums of cumulative infections and cumulative deaths at 360,000 and 6000, respectively, because the calibrated data for corresponding variables (i.e., cumulative infections and cumulative deaths) were the same for all scenarios below the chosen minimums.

**3.3.1 Impacts of increasing mask-wearing adherence level.** Fig 9 illustrates the impacts of vaccine efficacy and adherence level of mask-wearing on the COVID-19 spread over the next year. With different adherence levels for mask-wearing, while keeping other factors fixed, the cumulative infections at the end of the simulation period (i.e., Nov 5, 2022) could range from 393,304 (adherence level 70%)(S41) to 661,696 (adherence level 48%). The cumulative

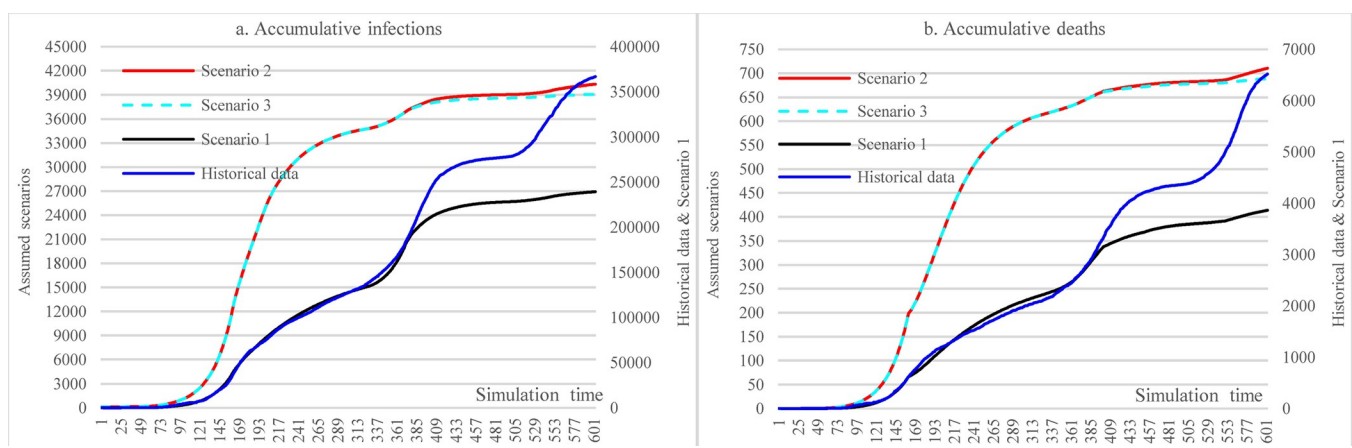

**Fig 8. Impacts of imposing social distancing on major social events and routine life activities in counterfactual analysis.**

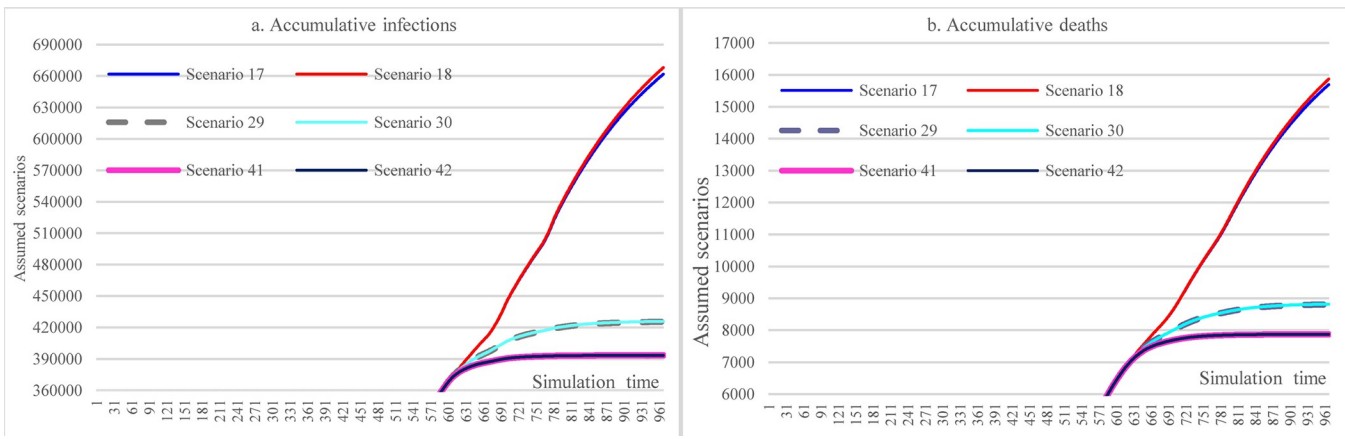

**Fig 9. Impacts of increasing adherence level of mask-wearing in scenario analysis.**

deaths were 15,697 in S17 (mask-wearing 48%) and 7867 in S41 (mask-wearing 70%), which led to a reduction of 49.88%. As for the impact of efficacy, the higher the mask-wearing adherence level, the lower the impact of reduced vaccine efficacy on the increase of cumulative infections was, which ranged from 0.013% to 0.96%.

**3.3.2 Impacts of the immunological period under low vaccination conditions.** Fig 10 demonstrates the impacts of the immunological period on cumulative infections and cumulative deaths. Similar to the pattern in Fig 9, the higher the mask-wearing adherence level, the lower the impact of the reduced immunological period on the increase of accumulative infections was, which ranged from 0.01% to 0.84%.

**3.3.3 Impacts of increasing stringencies of social distancing measures.** Under the assumption of a 48% mask-wearing adherence level, this group of scenarios compares the impacts of different stringency levels of SDMs on the spread dynamics of COVID-19 (refer to Fig 11). With the most stringent measure in SDMs, the cumulative infections could reach 424,681 at the end of the simulation (S25), which was 31,377 more than what could have been achieved in the case of 70% mask-wearing adherence level while having laxly implemented SDMs (S41). The cumulative deaths were 8823 in S25 (the most stringent measure in SDMs), which had a 43.79% reduction compared with S17.

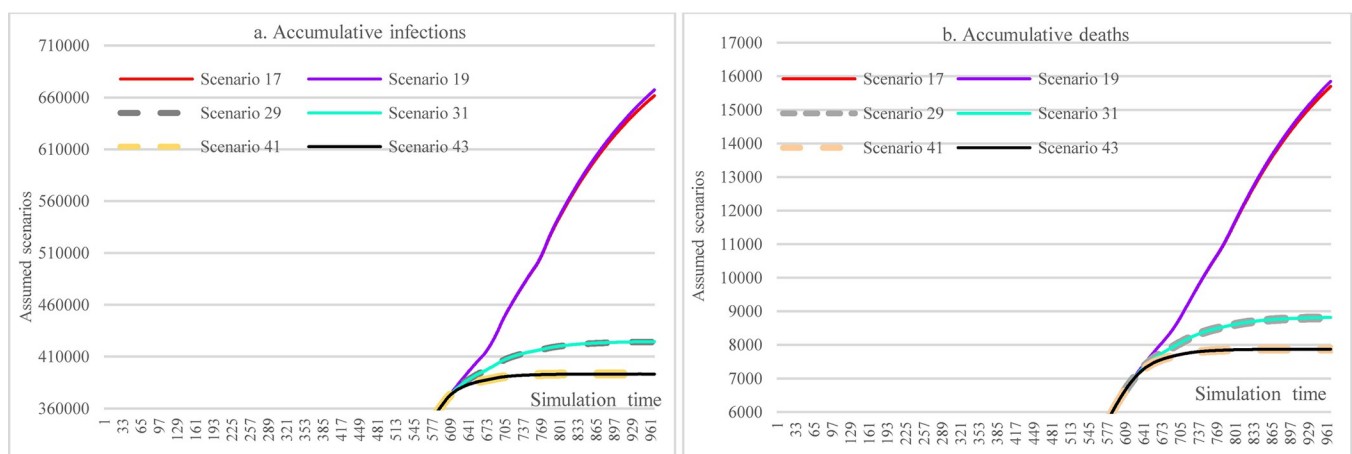

**Fig 10. Impacts of the immunological period on cumulative infection and cumulative deaths in scenario analysis (48% mask-wearing).**

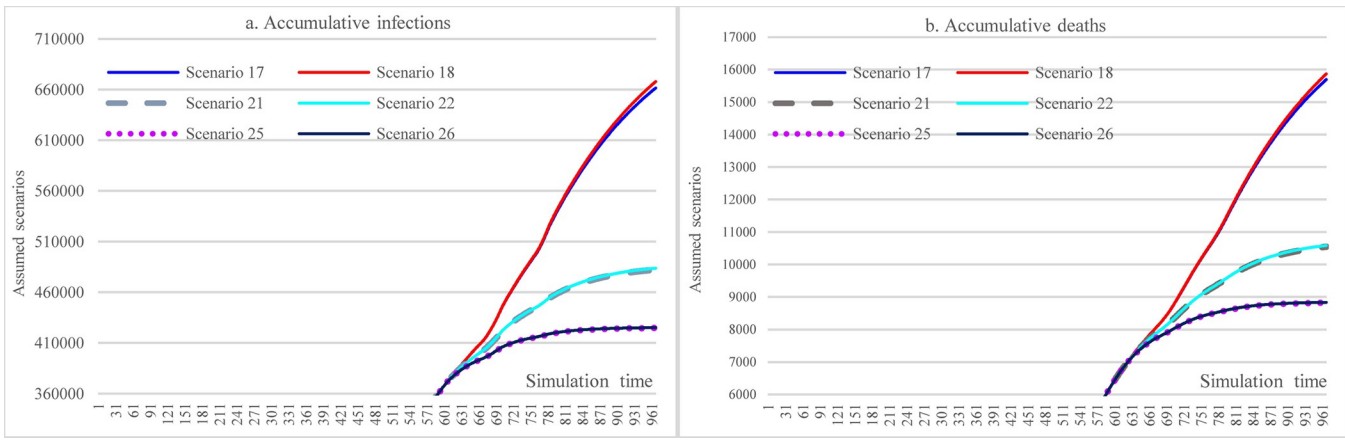

**Fig 11. Impacts of increasing stringency of social distancing measures in scenario analysis.**

**3.3.4 Impacts of hospital capacity.** This scenario group shows the impacts of different hospital capacities on the cumulative infections and cumulative deaths over time. Since the population size is very large and the results were very sensitive to the change of some conditions in our assumptions, the cumulative infections (S173 & S245) would exceed the number of total populations (because of repetitive infections) if the hospital capacity were smaller than 60% of the current level. These very large results just provide a reference here, which will not be used to compare with other variables. Changes in hospital capacity did not cause different accumulative infections (S17, S53, S137) as long as the hospital capacity was larger than 80% of the current one since the daily infections do not exceed hospital capacities. Compared to S137, the reductions in cumulative infections and cumulative deaths in S209 (11.89% and 13.88%) caused by increasing vaccine supply and vaccination capacity were noticeable.

**3.3.5 Impacts of combined NPIs and PIs.** Figs 12–14 show the impacts of the immunological period and vaccine efficacy under different mask-wearing adherence levels and levels of vaccination across the nation on the spread of COVID-19 in Ethiopia, which were calculated by taking the difference for the cumulative infections and cumulative deaths between two corresponding scenarios (e.g., S68-67). By comparing Figs 12 and 13, we can see that reduced vaccine efficacy produced larger effects than the lowered immunological period in causing

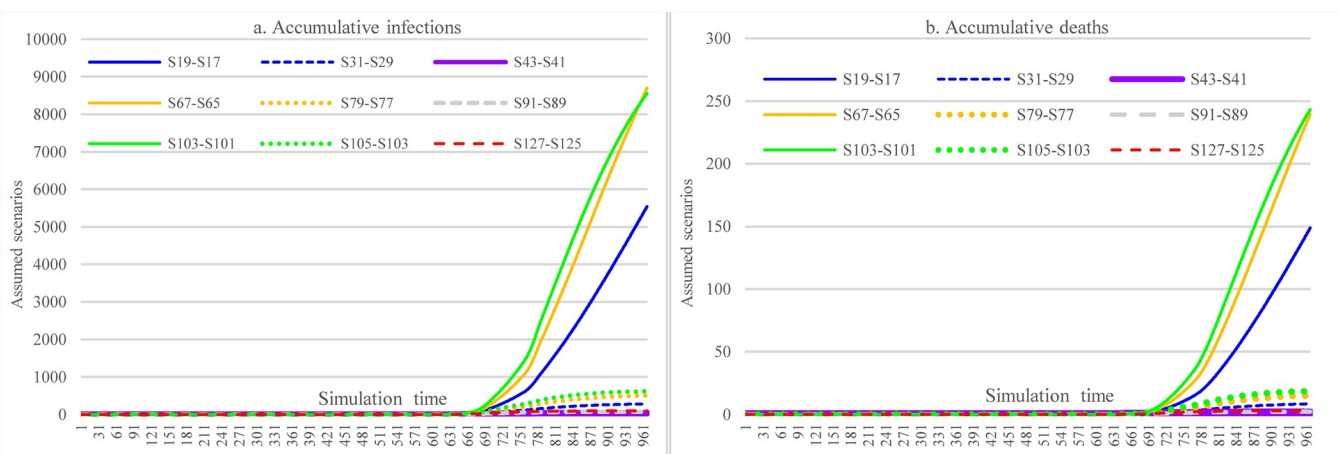

**Fig 12. Impacts of the immunological period on COVID-19 considering mask-wearing adherence and increasing vaccination capacity.**

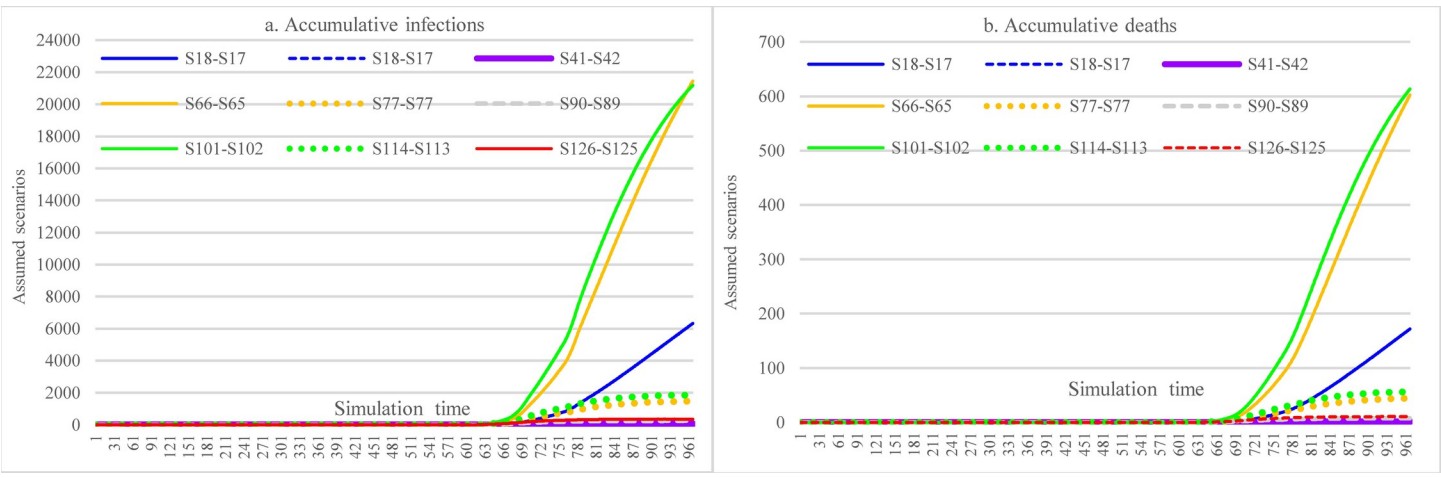

**Fig 13. Impacts of vaccine efficacy on COVID-19 spread considering mask-wearing adherence and increasing vaccination capacity.**

increased cumulative infections and cumulative deaths. The effect was especially obvious for the situation of higher vaccination rate (i.e., 30% population vaccinated 1st dose, 20% population vaccinated 2nd dose) and low mask-wearing adherence level (48%), which had the ratio of (S102-S101)/S101:(S103-S101)/S101 = 3.99%:1.61% (21181 vs. 8556 in number). Furthermore, in the situation mentioned above, the combined effects of both reduced immunological period and reduced vaccine efficacy could cause 5.6% (29737) more infections (Fig 14). However, the adverse effects of reduced immunological period and vaccine efficacy were to a large extent offset by increased mask-wearing adherence level, which is changed to the ratio of (S126-S125)/S101:(S127-S125)/S101 = 0.09%:0.025% (352 vs. 96 in number) with mask-wearing adherence level of 70%.

In Fig 15, S133 is the scenario having minimum cumulative infections and cumulative deaths (384094 and 7586) among all scenarios with different assumed interventions, where the interventions include: 30% and 20% of the total population being administered 1st and 2nd doses of COVID-19 vaccine, respectively, keeping 70% mask-wearing adherence level, adopting the most stringent SDMs, and holding vaccine efficacy and immunological period and hospital capacity unchanged. The maximum cumulative infections and cumulative deaths

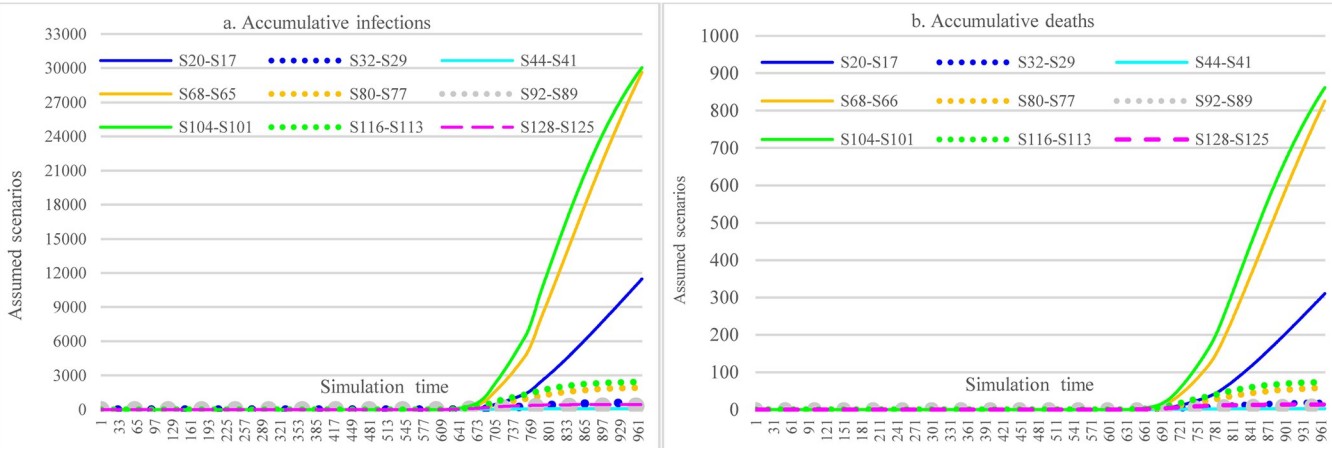

**Fig 14. Impacts of immunological period and vaccine efficacy on COVID-19 considering mask-wearing adherence and increasing vaccination capacity.**

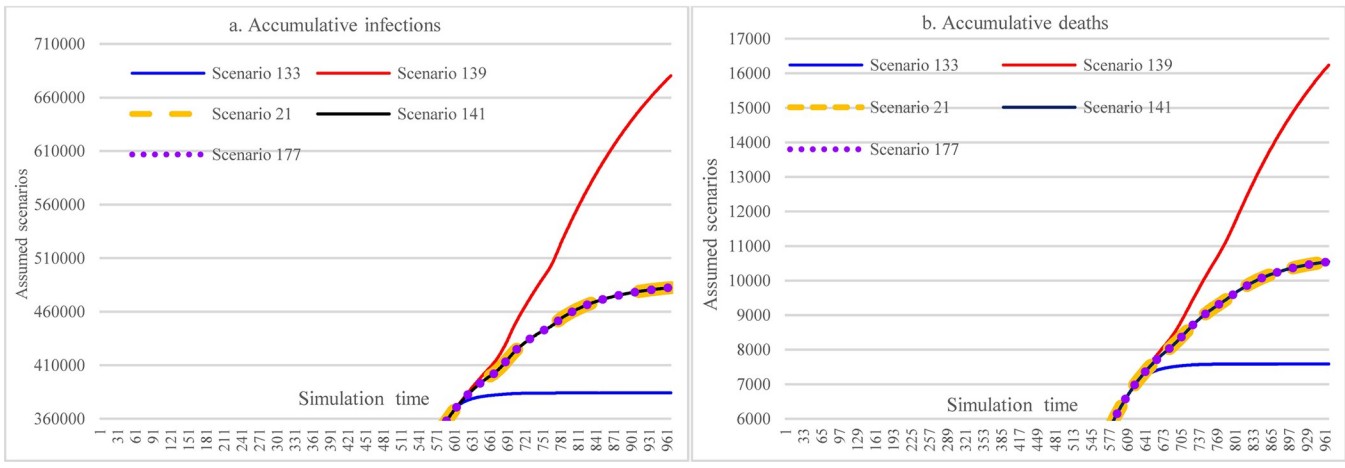

**Fig 15. The best-medium-worst outcomes among the defined scenarios.**

(680231 and 16235) happened in S139 where the interventions contain: reducing hospital capacity to 80% of the original level, laxly implementing SDMs, assuming vaccine supplying and administering pace calculated in historical data, keeping 48% mask-wearing adherence level, and adopting reduced vaccine efficacy (from 68.4% and 80% to 40% and 60% for the 1st and 2nd doses vaccine, respectively), and shortened immunological period (from 240 to 180 days). The maximum and minimum values differences were 77.10% and 114% of minimum cumulative infections and cumulative deaths, respectively. Scenarios S21, S141, and S177 represented the medium level of cumulative infections and cumulative deaths (482358 and 10549, please refer to C1, C4, and C5 for their corresponding settings.).

**3.3.6 Offsetting effects among multiple interventions.** In Fig 16, the results show that moderately increasing mask-wearing adherence level in the case of lax implementation of SDMs can achieve approximately the same effect (S25 vs. S29 and S37 vs. S41) as the situation of implementing stringent SDMs while having a low adherence level of mask-wearing (or due to insufficient supply of mask).

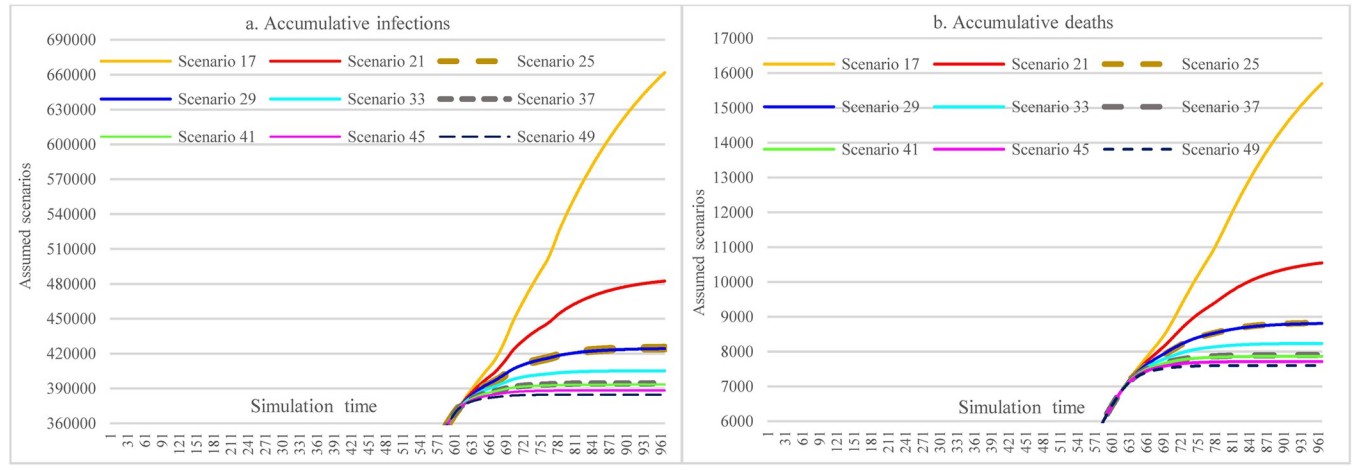

**Fig 16. Offsetting less stringent social distancing measures with increasing mask-wearing adherence level.**

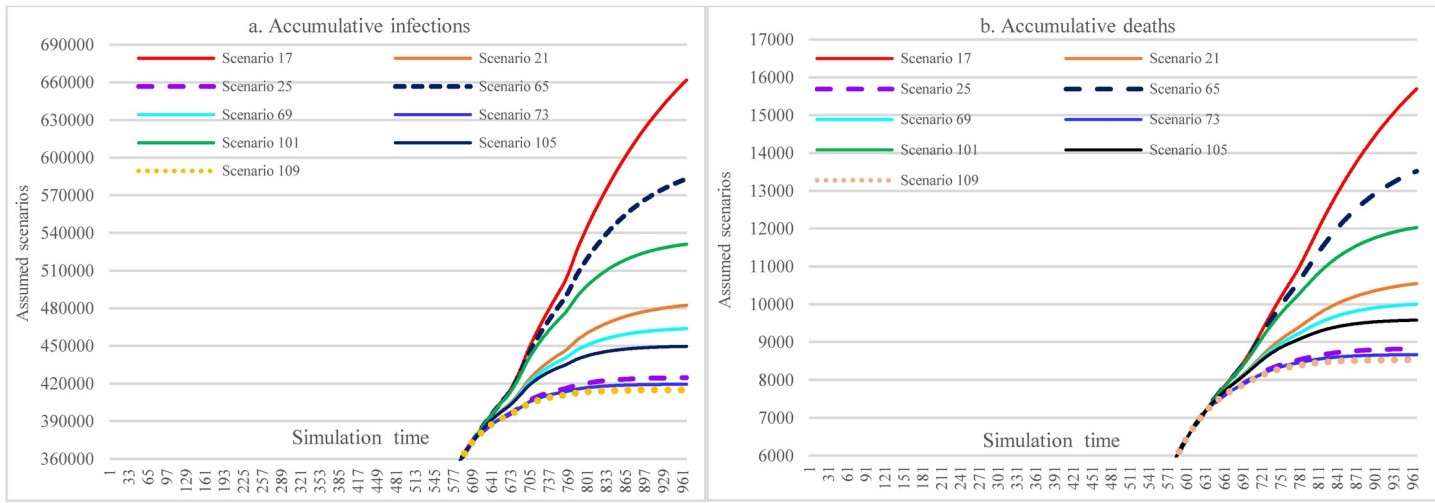

**Fig 17. Impacts of changing vaccination levels and implementing social distancing measures with mask-wearing adherence level of 48%.**

Figs 12–14 show the impacts of increasing vaccination and implementing SDMs on the cumulative infections and cumulative deaths under three mask-wearing adherence levels-i.e., 48%, 60%, and 70%. The impact of increasing vaccination to the extent of administering 1st dose for 30% population and 2nd dose for 20% of the population cannot achieve the effect that is achieved by increasing mask-wearing adherence level to 70% (showed in Fig 16). Furthermore, the effect was smaller than that of implementing more stringent SDMs. The impacts of increasing vaccination were lessened with more stringent SDMs and increasing the adherence level of mask-wearing because the cumulative infections and cumulative deaths were significantly reduced.

Fig 20, Fig 21, Fig 22 show the effect comparisons between implementing stringent SDMs and the scenarios of combining less stringent SDMs while increasing vaccination levels in the whole population of Ethiopia under different mask-wearing adherence levels-i.e., 48%, 60%, and 70%. As in the case of mask-wearing adherence level of 48%, implementing the most stringent SDMs (S25) had cumulative infections of 424,681 and cumulative deaths of 8,823. And

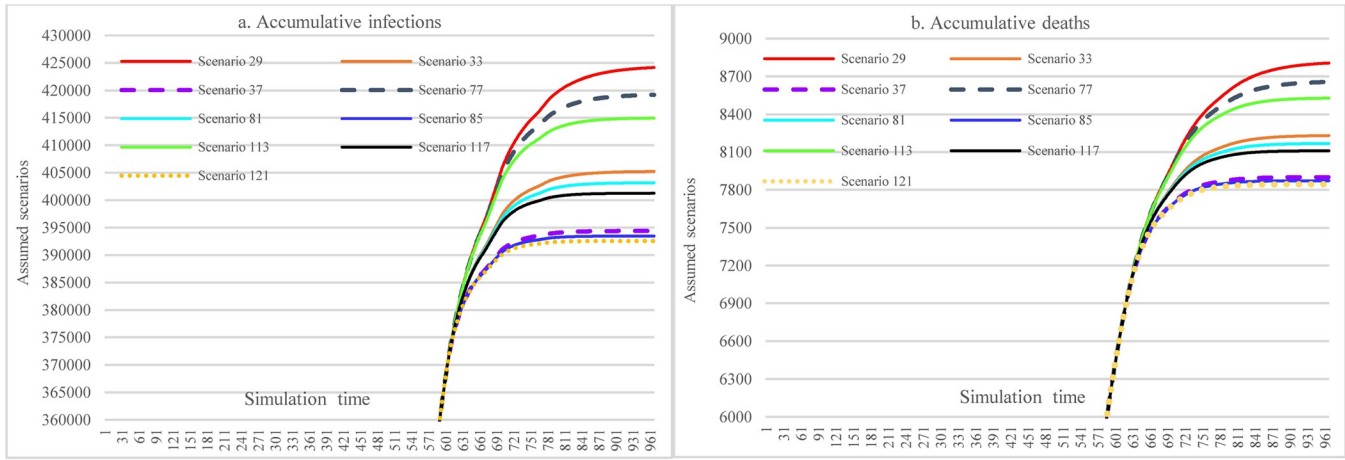

**Fig 18. Impacts of changing vaccination levels and implementing social distancing measures with mask-wearing adherence level of 60%.**

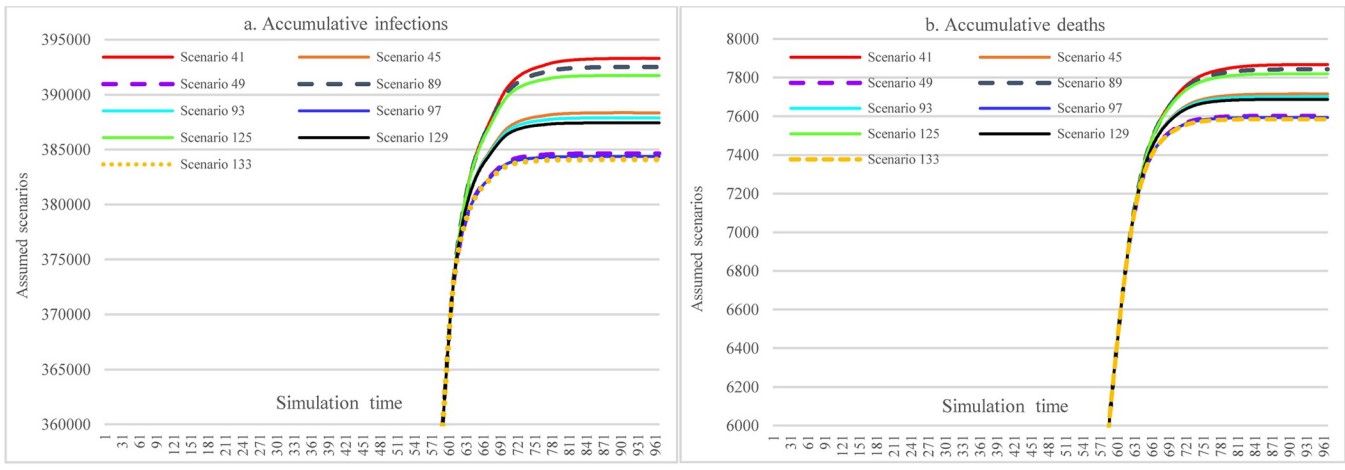

**Fig 19. Impacts of changing vaccination levels and implementing social distancing measures with mask-wearing adherence level of 70%.**

the scenario of vaccinating 30% and 20% total population with 1st and 2nd doses, respectively, and implementing the least stringent SDMs (for both major events and routine activities) (S101) had cumulative infections of 531083 and cumulative deaths of 12027, which were 25.05% and 36.31% more than of the S25. When considering the factor of reduced vaccine efficacy and immunological period, the differences were enlarged to be 32.13 and 46.07%, respectively. When the vaccination coverage was reduced to administering 1st and 2nd doses for 20% and 10% of the total population, respectively (S65), the cumulative infections and cumulative deaths of S65 were 37.29% and 53.21% more than that of S25. When considering the reductions in vaccine efficacy and immunological period, the differences were enlarged to 44.27% and 62.54% (S68 vs. S25). Under the assumption of a mask-wearing adherence level of 70%, implementing the most stringent SDMs (S49) had cumulative infections of 384,644 and cumulative deaths of 7,062. In comparison, the scenario of vaccinating 30% and 20% total population with 1st and 2nd doses, respectively, and implementing the least stringent SDMs (S125) had the cumulative infections of 391,735 and cumulative deaths of 7,819, whereas the latter (S125) were 1.84% and 2.86% more than of the former (S49). As for the vaccination cover of 20% and 10% of the total population having 1st and 2nd doses, respectively (S89), the

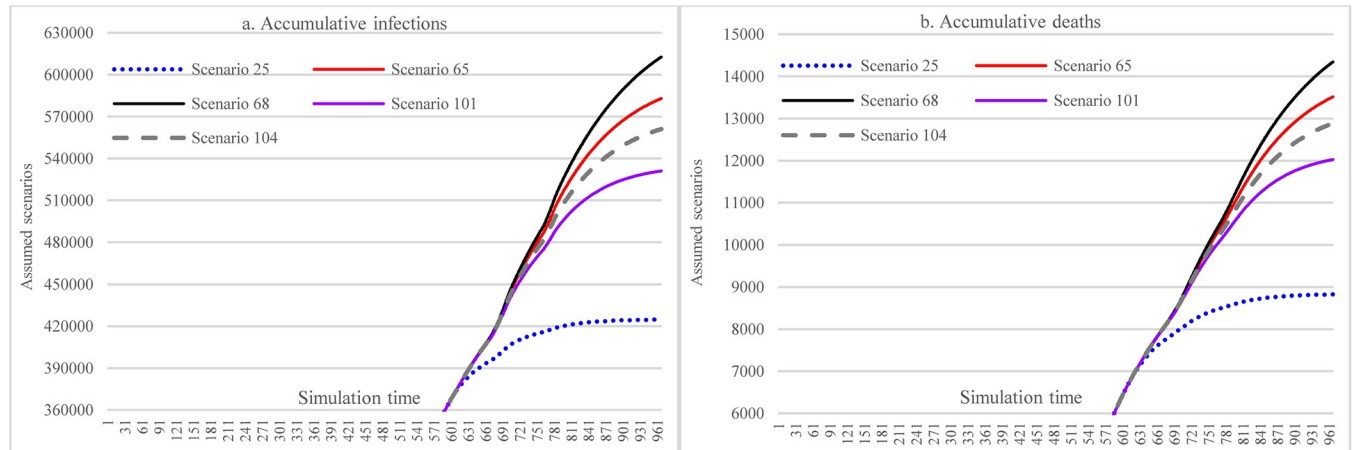

**Fig 20. Implementing stringent SDMs versus less stringent SDMs and increased vaccination levels under the mask-wearing adherence level of 48%.**

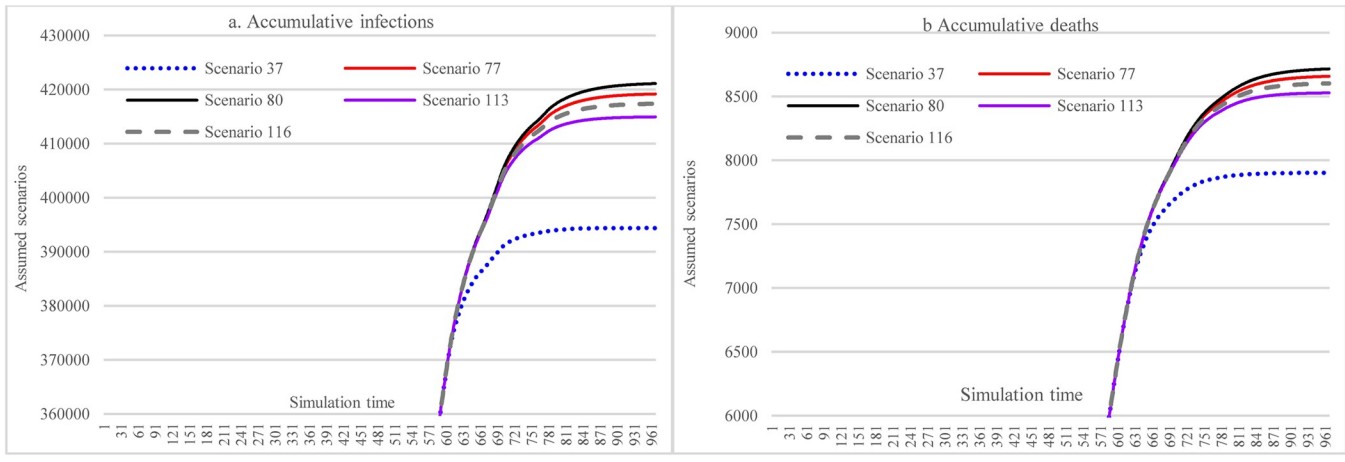

**Fig 21. Implementing stringent SDMs versus less stringent SDMs and increased vaccination levels under the mask-wearing adherence level of 60%.**

cumulative infections and cumulative deaths of S89 were 2.04% and 3.17% more than that of S49. Our study did not propose a higher vaccination level given that the issues in vaccine acquisition in Ethiopia cannot be overcome shortly.

In Fig 23, by comparing scenarios S37, S89, and S125, we found that increasing the vaccination rate and adherence level of mask-wearing under the circumstance of less stringent SDMs can achieve similar control results as that of the measure of lower-level vaccination rate and mask-wearing adherence with more stringent SDMs. Comparing scenarios S25 and S77 can further approve the observation mentioned above.

## 4 Discussion and conclusion

This SD model served as a framework for understanding the issues and gaps in the containment measures against COVID-19 in the past period and the spread dynamics of the infectious disease over the next year under different interventions and their combinations. This study simulated the results of 280 scenarios considering the vaccination level, efficacies of both 1$^{st}$ and 2$^{nd}$ vaccines, the immunological period for both vaccinated people and infected patients,

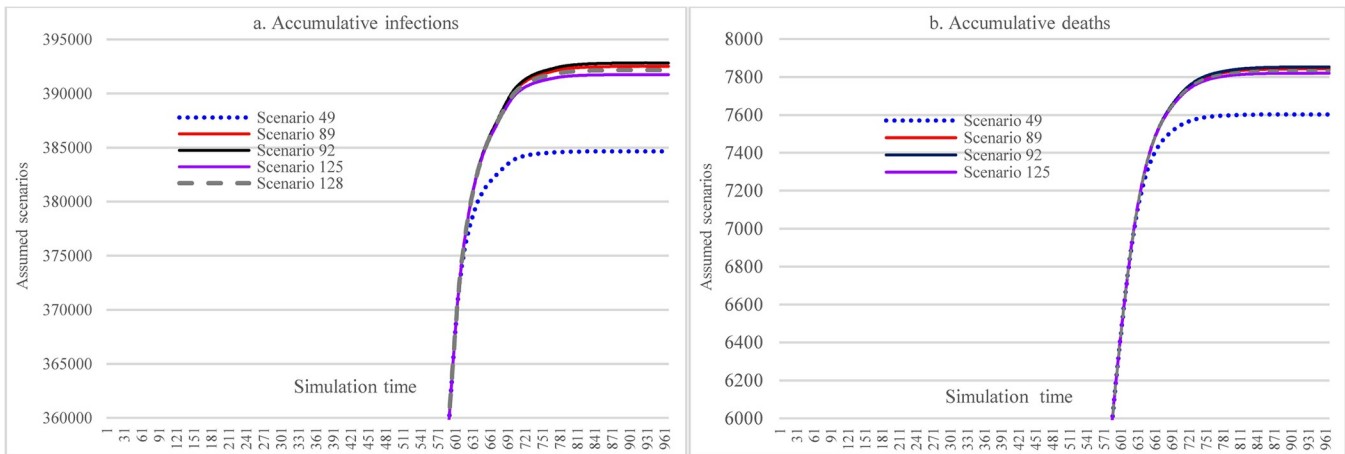

**Fig 22. Implementing stringent SDMs versus less stringent SDMs and increased vaccination levels under the mask-wearing adherence level of 70%.**

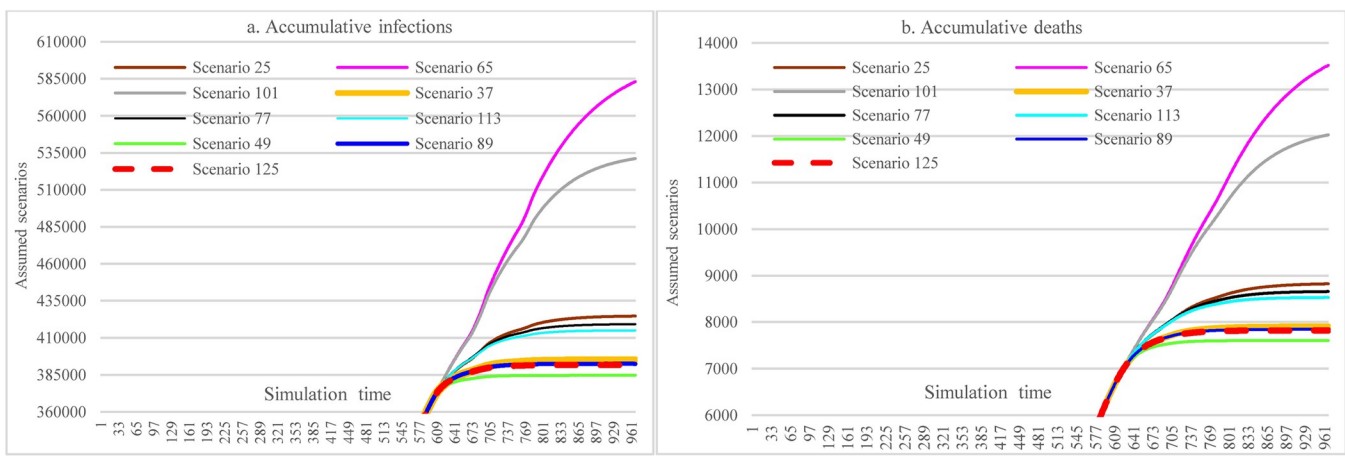

**Fig 23. Increasing vaccination levels and mask-wearing adherence as compensation for the less stringent SDMs.**

stringency levels in enforcing social distancing, adherence level of face mask-wearing, and hospital capacity. With evaluations and comparisons for alternative interventions, the results can inform policy and implementation science regarding the path, scale, duration, and stringency level of different interventions and their optimal combinations.

The fast spread of COVID-19 in Ethiopia can be attributed to joint effects of multiple factors including but are not limited to: (1) low adherence to face mask-wearing and hand hygiene practices; (2) failure to comply with social distancing; (3) most religious people attending more than three days a week in churches to pray together; (4) special religion holidays celebration events were held for more than 7 days across the country by Orthodox Christian; (5) more than three holidays events by Muslim followers; (6) extensive protests in a different state in 2020 during a pandemic; (7) massive rally during the country election on 21 June 2021 [6, 8–10]; and (8) the war between the federal government and Tigray Region since November 4, 2020, which not only increased COVID-19 infection risk of fighters and almost two million displaced migrant but also disrupted the vaccine supply and administration and everyday operations of many hospitals [11].

In the counterfactual analysis, we found that keeping high mask-wearing adherence since the outbreak of COVID-19 in Ethiopia could have significantly reduced the infection under the condition of low vaccination level or unavailability of vaccine (Fig 7). In the trend analysis, higher mask-wearing adherence still played the dominant role in significantly reducing infection, and the best outcome was attained with one condition being a 70% adherence level. In terms of the social distancing measures, reducing or canceling major social events (e.g., religious gatherings and protests) can achieve a better outcome than imposing the same constraints on people's routine life activities. Moreover, since the daily infections did not exceed hospital treatment capacity in the early stage of the COVID-19 pandemic, it seemed that increasing hospital capacity did not play a pivotal role in significantly reducing infections. However, the significant contributions of hospitalizing the COVID-19 infected patient were to reduce the death toll caused by severe cases and carry out necessary quarantine functions, which significantly reduced the sources of further transmission [104, 105]. This can be testified in Fig 24, where the insufficient capacity for hospitals (including quarantine hospitals) could lead to a considerable number of accumulative infections.

In the trend analysis, under the conditions of low vaccination rate, reduced vaccine efficacy, and immunological period had no apparent effects on accumulative infections. Furthermore,

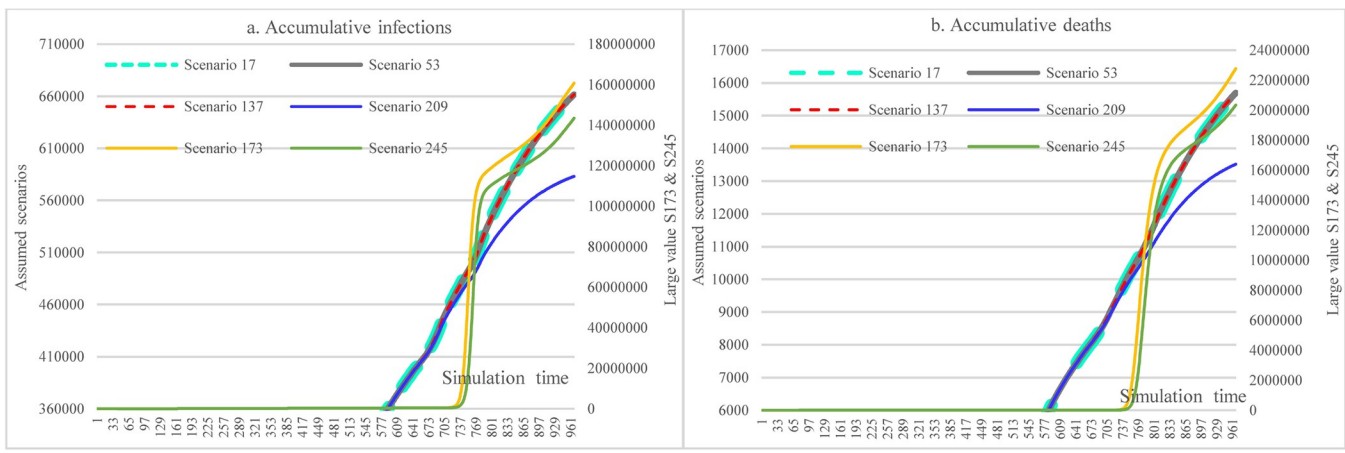

**Fig 24. Impacts of hospital capacity in scenario analysis.**

increasing mask-wearing adherence and enforcing more stringent social distancing were two main measures that could significantly reduce possible infections. Higher mask-wearing adherence had more significant impacts than enforcing social distancing measures in our settings. As the vaccination rate increases, factors such as vaccine efficacy and immunological period that affect vaccination effectiveness started to take effect, where the reduced efficacy could cause more infections than that of shortened immunological periods. Offsetting effects of multiple interventions (strengthening one or more interventions while loosening others) could be applied when the levels or stringencies of one or more interventions needed to be adjusted for catering to particular needs (e.g., less stringent SDMs to reboot the economy or cushion insufficient resources in some areas).

Like all other research, the current study has certain limitations. First of all, since the number of possible scenarios by combining multiple interventions and their different levels can be very large, this paper only picked and analyzed minimal representative scenarios. Secondly, data used in the simulation were at the aggregate level without incorporating regional heterogeneity. Moreover, the ever-mutating COVID-19 virus means that current analysis might not be able to reflect future pandemic dynamics.

Several possible studies can be conducted in the future research: (1) the model can be easily extended to evaluate impacts of containment measures for emerging infectious diseases; (2) by adding a demand-supply interaction substructure for personal protection equipment (PPE) such as face mask, the revised model can be used to evaluate impact of PPE logistics on the spread dynamics of COVID-19 (considering adherence level and protection ability); (3) this model can help find the optimal combination of containment measures; (4) the model can be expanded to inform decision makers of determining optimal lock-down window and period by jointly considering the supply & demand dynamics of PPE and vaccine, vaccine efficacy and immunological period, effect of social distancing measures, and characteristics of the mutants of the disease; (5) a revised model can also help undertake hospital capacity planning for dealing with public health emergency like COVID-19 pandemic; and (6) the model can also be used to define the necessary vaccination level and consequently the immune barrier in a given period by considering vaccine efficacy and immunological period for both vaccinated and infected people.

In conclusion, SD can be a handy tool for expedited learning for designing and implementing public health emergency policies (or interventions), especially those involving multiple

interventions that could have thousands of possible implementation paths. It can help pinpoint issues and gaps in the historical path and investigate and choose the appropriate path for achieving better containment and control outcomes given limited resources, complicated socioeconomic systems, and characteristics of the emerging infectious disease. This evaluation of the NPIs and PIs will help provide constructive inputs to inform policy and decision-making regarding COVID-19 and other emerging infectious diseases.

## Supporting information

**S1 Appendix.**
(DOCX)

**S1 Data.**
(ZIP)

## Author Contributions

**Conceptualization:** Shiyong Liu, Weiwei Zhang, Peng Jia.

**Data curation:** Hongli Zhu, Haimanote Belay.

**Formal analysis:** Wenwen Zheng, Haimanote Belay, Weiwei Zhang.

**Funding acquisition:** Shiyong Liu.

**Investigation:** Peng Jia.

**Methodology:** Shiyong Liu, Weiwei Zhang.

**Resources:** Wenwen Zheng.

**Supervision:** Shiyong Liu.

**Validation:** Hongli Zhu, Shiyong Liu, Tadesse Guadu Delele.

**Writing – original draft:** Hongli Zhu, Shiyong Liu.

**Writing – review & editing:** Shiyong Liu, Ying Qian, Yirong Wu, Peng Jia.

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
