## [Decision Letter · Decision Letter 0]

18 Apr 2022

PONE-D-22-02713Assessing the dynamic impacts of non-pharmaceutical and pharmaceutical intervention measures on the containment results against COVID-19 in EthiopiaPLOS ONE

Dear Dr. Liu,

Thank you for submitting your manuscript to PLOS ONE. After careful consideration, we feel that it has merit but does not fully meet PLOS ONE’s publication criteria as it currently stands. Therefore, we invite you to submit a revised version of the manuscript that addresses the points raised during the review process. Please consider the comments pointed out by the reviewer. Also, analyze and discuss the potential influence of SARS-CoV-2 variants.

We look forward to receiving your revised manuscript.

Kind regards,

Vinícius Silva Belo

Academic Editor

PLOS ONE

Journal Requirements:

“The authors declare no conflict of interests”

Reviewers' comments:

Reviewer's Responses to Questions

**Comments to the Author**

1. Is the manuscript technically sound, and do the data support the conclusions?

Reviewer #1: Partly

2. Has the statistical analysis been performed appropriately and rigorously? 

Reviewer #1: No

3. Have the authors made all data underlying the findings in their manuscript fully available?

Reviewer #1: No

4. Is the manuscript presented in an intelligible fashion and written in standard English?

Reviewer #1: No

5. Review Comments to the Author

Reviewer #1: Comments to Authors

Currently, COVID-19 is a major public health issue and assessing the dynamic impacts of non-pharmaceutical and pharmaceutical intervention measures against COVID-19 is an important area of study. The authors try to investigate system dynamics model to capture COVID-19 characteristics, major social events, stringencies of containment measures, and vaccine dynamics. To improve the quality of the paper, I suggest to address the following issues:

Abstract

- Avoid abbreviation from the abstract. i.e. SD, SDM

Background

- The background section didn’t intensively explore literatures related with this study.

- Many statements written without citing references that support the statement. Such us line 64-66, line 72-78, Line

- This section didn’t motivate why doing this study is important and what are the limitation of other similar studies, i.e. Ejigu et al (2021)

- Need proof reading. There are many grammatical issues across the paper.

i.e., line 61 “only were 1.4%....”.

line 161 “This study is to better understand…”

Method

- Line 161 is not a meaningful sentence. “This study is to better understand…”

- Motivation to the considered “Model structure and formulations” were not clearly stated. Under what assumption this model is valid? How you get the model parameter input parameters?

- Add links/hyperlinks (cite websites) from where you got the data under the “Data collection and quality” subsection.

- Further in this section, you need to explain

a) How the Immunological period (days) 240 and 180 determined.

b) How the vaccine efficacy determined

c) Briefly explain how other parameters also fixed

-

Result and discussion

- Many of the assumptions set for the scenarios are far away the reality. For example, line 272-273, it was stated that “.. adherence level in wearing face was assumed 55%” which is really far away from the reality in the context of Ethiopia. As far as I know, 75% of the population in Ethiopia are living in rural areas and majority of them didn’t her about facemask. Further, in line 295 “…. adherence level of mask-wearing (48%, 60%, and 70%).”

- Revise the assumed adherence level related with face masking, social distancing,… in the context of Ethiopia.

- Line 297 “…total of 968 days”, but in line 205 “… total of 603 days” which one is correct.

- Finding of other studies related with this work were not well discussed.

- Limitation of the study not clearly stated. i.e., the quality of the data and others not clearly addressed.

6. PLOS authors have the option to publish the peer review history of their article (what does this mean?). If published, this will include your full peer review and any attached files.

Reviewer #1: No

---

## [Author Response · Author response to Decision Letter 0]

13 Jun 2022

First of all, we would like to sincerely thank the editor and anonymous reviewers for their comments and constructive critiques. We appreciate the opportunity to address the comments by submitting a revision. We have considered all comments with care, and we believe the revisions address these comments and have greatly improved the paper. In this revision report, we explain the responses to each comment in detail. We have reproduced the original comments in black and typed our responses in blue for clarity. 

Responses to Reviewer #1

1. - Avoid abbreviation from the abstract. i.e. SD, SDM

Response:

Thank you for mentioning this point. We have changed those abbreviations in the abstract to their full names. 

2. -The background section didn’t intensively explore literatures related with this study.

Response:

We appreciate that you pointed this out. We further searched relevant studies and summarized them in the revised manuscript. Relevant references are listed below. 

References: 

1. Ayele W, Tesfaye L, Abagero A, Taye G, Abate B, Habtamu T, et al. COVID 19 Epidemic trajectory modeling results for Ethiopia. Ethiop J Heal Dev. 2021;35: 25–32.

2. Ayele TA, Fentahun N, Tiruneh SA, Haile M, Zeru T, Meseret G, et al. Spatial variation and level of compliance on COVID-19 Prevention strategies in Amhara region, Ethiopia: Observational survey. Ethiop J Heal Dev. 2021;35.

3. Endriyas M, Kawza A, Alano A, Hussen M, Shibru E. COVID-19 prevention practices in urban setting during early introduction of the disease: results from community survey in SNNP Region, Ethiopia. BMJ Open. 2021;11. doi:10.1136/bmjopen-2020-047373

4. Haftom M, Petrucka PM. Determinants of Face Mask Utilization to Prevent Covid-19 Pandemic among Quarantined Adults in Tigrai Region, Northern Ethiopia, 2020. Clin Nurs Res. 2021;30: 1107–1112. doi:10.1177/10547738211013219

5. Tucho GT, Kumsa DM. Universal Use of Face Masks and Related Challenges During COVID-19 in developing countries. Risk Manag Healthc Policy. 2021;14: 511–517. doi:10.2147/RMHP.S298687

6. Zewude B, Melese B, Addis E, Solomon W. Changing patterns of compliance with protective behavioral recommendations in the post first-round COVID-19 vaccine period among healthcare workers in Southern Ethiopia. Risk Manag Healthc Policy. 2021;14: 3575–3587. doi:10.2147/RMHP.S325699

7. Bushira KM. Modeling the effectiveness of social distancing interventions on the epidemic curve of coronavirus disease in Ethiopia. Model earth Syst Environ. 2021; 1–11. doi:10.1007/s40808-021-01190-9

8. Deressa W, Worku A, Abebe W, et al. Social distancing and preventive practices of government employees in response to COVID-19 in Ethiopia. PLOS ONE. 2021; 16(9): e0257112:1-22. https://doi.org/10.1371/journal.pone.0257112

9. Fikrie A, Amaje E, Golicha W. Social distancing practice and associated factors in response to COVID-19 pandemic at West Guji Zone, Southern Ethiopia, 2021: A community based cross-sectional study. PLoS One. 2021;16. doi:10.1371/journal.pone.0261065

10. Hailu W, Derseh L, Hunegnaw MT, Tesfaye T, Angaw DA. Compliance, barriers, and facilitators to social distancing measures for prevention of coronavirus disease 2019 in Northwest Ethiopia, 2020. Curr Ther Res Exp. 2021;94. doi:10.1016/j.curtheres.2021.100632

11. Tolu LB, Ezeh A, Feyissa GT. How prepared is Africa for the COVID-19 pandemic response? The case of Ethiopia. Risk Manag Healthc Policy. 2020;13: 771–776. doi:10.2147/RMHP.S258273

3. - Many statements written without citing references that support the statement. Such us line 64-66, line 72-78, Line

Response:

Thank you for raising this point. We have double-checked the whole paper and added citations at relevant places. 

4. - This section didn’t motivate why doing this study is important and what are the limitation of other similar studies, i.e. Ejigu et al (2021)

Response:

Thank you for the suggestions. We have added a paragraph deliberating this point. 

5. - Need proof reading. There are many grammatical issues across the paper. i.e., line 61 “only were 1.4%....”. line 161 “This study is to better understand…”

Responses: 

Many thanks to the reviewer for indicating this situation. We agree that improving the writing is very important for guaranteeing the quality of the manuscript. We have invited professional English-editing service to edit and polish the manuscripts. 

6. - Line 161 is not a meaningful sentence. “This study is to better understand…”

Responses: 

Thank you for helping improve the writing of this manuscript, we have revised this sentence and checked other places as well. 

7. - Motivation to the considered “Model structure and formulations” were not clearly stated. Under what assumption this model is valid? How you get the model parameter input parameters?

Responses: 

Thank you for mentioning this. We did have some elaboration on the motivation for using the extended SEIR model in “Introduction” and subsection of “Model structure and formulations” Built on previous research, our extended SEIR model tends to include not only the state variables of susceptible population, exposed population, symptomatic patients, asymptomatic patients, mild cases, severe cases, deaths, recovered cases, but also the sub-structure of hospitalized patients, un-hospitalized patients, vaccine administration. 

Piecewise calibration was conducted to estimate necessary model parameters, given that the cumulative infections and cumulative deaths did not follow an expectedly exponential growth curve. Historical data were split into two parts with one part being used for building the model. And the model was then verified and validated by the other part of the data. As for the model parameters, some of the parameters were obtained through the fitting with historical and they were verified through repetitive calibrations. Besides, the value settings for other parameters were done by referring to literature, which were clearly presented in Table 1. 

8. - Add links/hyperlinks (cite websites) from where you got the data under the “Data collection and quality” subsection.

Responses: 

Thank you for advising on this point, we have put the link in the relevant place of the “Data collection and quality” subsection. 

9. - - Further in this section, you need to explain

a) How the Immunological period (days) 240 and 180 determined.

b) How the vaccine efficacy determined 

c) Briefly explain how other parameters also fixed

Responses: 

Thank you for mentioning this. For a) and b), we have listed literature that indicated the source of the value ([1,2] for a) and [3-6] for b)). In this revision, we added several more references. Please also refer to the literature provided below. Regarding c), some of the parameters were obtained through the fitting with historical and they were verified through repetitive calibrations. Besides, the value settings for other parameters were done by referring to literature, which was clearly presented in Table 1. 

References:

1. Khoury DS, Cromer D, Reynaldi A, et al. Neutralizing antibody levels are highly predictive of immune protection from symptomatic SARS-CoV-2 infection. Nat Med 2021;27: 1205-11

2. Tenforde MW, Self WH, Naioti EA, et al. Sustained effectiveness of Pfizer BioNTech and Moderna vaccines against COVID-19 associated hospitalizations among adults-United States, March, July 2021. Morb Mortal Wkly Rep. 2021; 70(34), 1156.

3. Bernal JL, Andrews N, Gower C, et al. Effectiveness of the Pfizer-BioNTech and Oxford-AstraZeneca vaccines on COVID-19 related symptoms, hospital admissions, and mortality in older adults in England: test negative case-control study. BMJ-British Med J. 2021; 373:n1088. https://doi.org/10.1136/bmj.n1088

4. Dagan N, Barda N, Kepten E, et al. BNT162b2 mRNA Covid-19 vaccine in a nationwide mass vaccination setting. N Engl J Med. 2021; 384(15):1412-1423. https://doi.org/10.1056/NEJMoa2101765

5. Jara A, Undurraga EA, Gonzalez C, et al. Effectiveness of an inactivated SARS-CoV-2 Vaccine in Chile. N Engl J Med. 2021; 385(10):875-884. https://doi.org/10.1056/NEJMoa2107715

6. Shekhar R, Sheikh A, Upadhyay S, et al. COVID-19 vaccine acceptance among health care workers in the United States. Vaccines. 2021;9,119. https://doi.org/10.3390/vaccines9020119

10. - - - Many of the assumptions set for the scenarios are far away the reality. For example, line 272-273, it was stated that “.. adherence level in wearing face was assumed 55%” which is really far away from the reality in the context of Ethiopia. As far as I know, 75% of the population in Ethiopia are living in rural areas and majority of them didn’t her about facemask. Further, in line 295 “…. adherence level of mask-wearing (48%, 60%, and 70%).”

- Revise the assumed adherence level related with face masking, social distancing… in the context of Ethiopia.

Responses: 

Thanks to the reviewer for pointing this out. We agree that the compliance level to face mask-wearing in Ethiopia was very low, especially in those poor rural areas at the outset of COVID-19. However, with the supply of face masks, the level of compliance witnessed an obvious increase, which increased from 30% to as high as 62% (IHME). We also searched the literature and found results from some surveys done by Ethiopian scholars showing that the levels of adherence (Endriyas et al., 2021; Haftom and Petrucka, 2020) were more than 50%. Ayele Wondimu et al. evaluated the spread dynamics of COVID-19 in Ethiopia under the assumptions of face mask utilization of 20%, 40%, and 60%.Therefore, the percentages used in our scenarios represented relatively reasonable assumptions. 

References:

1. IHME, Mask use in Ethiopia. Accessed on June 5th, 2022 at 

https://covid19.healthdata.org/ethiopia?view=mask-use&tab=trend

2. Endriyas M, Kawza A, Alano A, Hussen M, Shibru E. COVID-19 prevention practices in urban setting during early introduction of the disease: Results from community survey in SNNP Region, Ethiopia. BMJ Open. 2021;11. doi:10.1136/bmjopen-2020-047373

3. Haftom M, Petrucka PM. Determinants of Face Mask Utilization to Prevent Covid-19 Pandemic among Quarantined Adults in Tigrai Region, Northern Ethiopia, 2020. Clin Nurs Res. 2021;30: 1107–1112. doi:10.1177/10547738211013219

4. Ayele W, Tesfaye L, Abagero A, Taye G, Abate B, Habtamu T, et al. COVID 19 Epidemic trajectory modeling results for Ethiopia. Ethiop J Heal Dev. 2021;35: 25–32.

11. - Line 297 “…total of 968 days”, but in line 205 “… total of 603 days” which one is correct.

Responses: 

Thank you for mentioning this point. Our study has two parts with one focusing on counterfactual analysis that has a total of 603 days and the other focusing on the prediction that covers one year (365 days). Therefore, the total simulation days considering prediction are 968 days. To reduce confusion, we have made changes at the corresponding place. 

12. - Finding of other studies related with this work were not well discussed.

Responses: 

Thank you so much for pointing this out. We have added relevant studies in the Introduction, Results, and Discussion parts. 

13. - - Limitation of the study not clearly stated. i.e., the quality of the data and others not clearly addressed.

Responses: 

Thanks for your suggestions. We have added limitations at the end of the discussion part.

---

## [Editor Report · Decision Letter 1]

27 Jun 2022

Assessing the dynamic impacts of non-pharmaceutical and pharmaceutical intervention measures on the containment results against COVID-19 in Ethiopia

PONE-D-22-02713R1

Dear Dr. Liu,

We’re pleased to inform you that your manuscript has been judged scientifically suitable for publication and will be formally accepted for publication once it meets all outstanding technical requirements.

Kind regards,

Vinícius Silva Belo

Academic Editor

PLOS ONE
---

## [Editor Report · Acceptance letter]

8 Jul 2022

PONE-D-22-02713R1 

Assessing the dynamic impacts of non-pharmaceutical and pharmaceutical intervention measures on the containment results against COVID-19 in Ethiopia 

Dear Dr. Liu:

I'm pleased to inform you that your manuscript has been deemed suitable for publication in PLOS ONE. Congratulations! Your manuscript is now with our production department. 

Kind regards, 

on behalf of

Dr. Vinícius Silva Belo 

Academic Editor

PLOS ONE